# A wind environment and Lorentz factors of tens explain gamma-ray bursts X-ray plateau

Hüsne Dereli-Bégué [1] ✉, Asaf Pe'er[1] ✉, Felix Ryde [2], Samantha R. Oates [3], Bing Zhang [4,5] & Maria G. Dainotti[6,7,8]

Gamma-ray bursts (GRBs) are known to have the most relativistic jets, with initial Lorentz factors in the order of a few hundreds. Many GRBs display an early X-ray light-curve plateau, which was not theoretically expected and therefore puzzled the community for many years. Here, we show that this observed signal is naturally obtained within the classical GRB fireball model, provided that the initial Lorentz factor is rather a few tens, and the expansion occurs into a medium-low density wind. The range of Lorentz factors in GRB jets is thus much wider than previously thought and bridges an observational gap between mildly relativistic jets inferred in active galactic nuclei, to highly relativistic jets deduced in few extreme GRBs. Furthermore, long GRB progenitors are either not Wolf-Rayet stars, or the wind properties during the final stellar evolution phase are different than at earlier times. Our model has predictions that can be tested to verify or reject it in the future, such as lack of GeV emission, lack of strong thermal component and long (few seconds) variability during the prompt phase characterizing plateau bursts.

Gamma-ray bursts (GRBs) are the most energetic explosions known in the Universe and are also known to have the most relativistic jets, with initial expansion Lorentz factors of $100 < \Gamma_i < 1000$[1–3]. One of the most puzzling results in the study of GRBs is the existence of a long plateau in the early X-ray light curve (up to thousands of seconds)[4–7] of a significant fraction of GRBs (43% until 2009[8] and 56% until 2019[7]). This plateau, not predicted theoretically[9], was first detected by the Neil Gehrels Swift Observatory[10] in 2005, and despite the long time passed since its discovery, its origin is still highly debated in the literature, with many authors suggesting various extensions to the classical fireball model[9] in order to explain it. Within the classical GRB fireball model[9,11,12], the huge amount of energy ($10^{51}$–$10^{54}$ ergs) released in a compact region results in the creation of an optically thick fireball. The fireball, made of baryons, $e^{\pm}$ and photons, is accelerated by its own radiative pressure to highly relativistic speeds. Following an initial

acceleration phase, the plasma coasts, cools and collects material from the circumstellar medium (CSM) which causes it to gradually slow. The observed signal is mainly due to synchrotron radiation from particles heated by the strong shocks that exist above the photosphere, and is predicted to gradually decay as the ejecta slows down in time, as is indeed observed (the so called afterglow).

To explain the plateau, the first and most commonly used idea is the continuous energy injection from a central compact object which can be a newly formed black hole[4,6,13] or a millisecond magnetar[14]. Other notable ideas include two components[15–17] or multi component[18] jet models; forward shock emission in homogeneous media[18]; scattering by dust/modification of ambient density by gamma-ray trigger[19,20]; dominant reverse shock emission[21,22]; evolving microphysical parameters[19]; and viewing angle effects in which jets are viewed off-axis[18,23,24]. While each of these ideas is capable of explaining

[1]Department of Physics, Bar-Ilan University, Ramat-Gan 52900, Israel. [2]Department of Physics, KTH Royal Institute of Technology and The Oskar Klein Centre, SE-106 91 Stockholm, Sweden. [3]School of Physics and Astronomy & Institute for Gravitational Wave Astronomy, University of Birmingham, Birmingham B15 2TT, UK. [4]Nevada Center for Astrophysics, University of Nevada, Las Vegas, NV 89154, USA. [5]Department of Physics and Astronomy, University of Nevada, Las Vegas, NV 89154, USA. [6]National Astronomical Observatory of Japan, 2-21-1 Osawa, Mitaka, Tokyo 181-8588, Japan. [7]The Graduate University for Advanced Studies, SOKENDAI, Shonankokusaimura, Hayama, Miura District, Kanagawa 240-0193, Japan. [8]Space Science Institute, Boulder, CO 80301, USA. ✉e-mail: husnedereli@gmail.com; asaf.peer@biu.ac.il

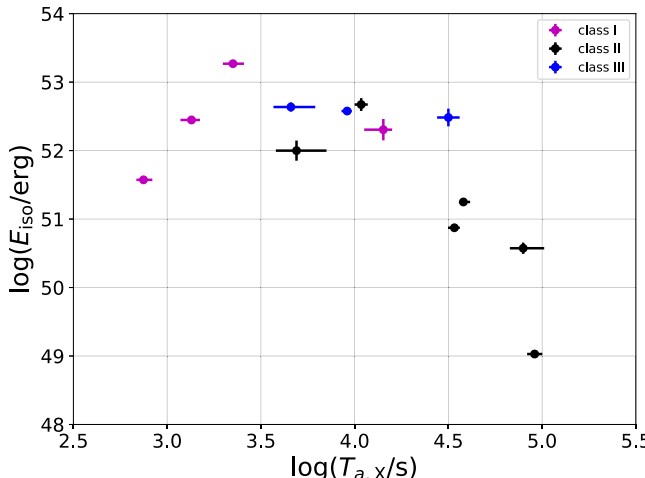

**Fig. 1 | Total Isotropic Energy, $E_{iso}$ as a function of the time at the end of the plateau phase, $T_{a,X}$ in the X-ray band.** Purple, black and blue points represent GRBs in the three different classes I, II, III respectively (see Supplementary Tables 1–3). The errors correspond to a significance of one sigma. Note that these three classes, selected only based on the temporal indices of their X-ray and optical afterglow light curves, are correlated with the prompt phase energy as well as the break time: each class occupies a different region in the $E_{iso}$ - $T_{a,X}$ parameter space. This fact provides a further, independent tool that increases our confidence in the selection criteria we use for these classes. The source data to reproduce this figure are provided as a Source Data file.

the observed plateau under certain conditions, they all require an external addition to the basic fireball model scenario, and in some cases they cannot address the full set of properties of the plateau phase (such as the flux, slope or duration). A thorough discussion on the advantages and weaknesses of each of these ideas appear in Ref. 25. We provide a short comparison with some of the recent proposed ideas in Supplementary Discussion, Comparison with other models aimed at explaining the X-ray plateau.

Plateaus are seen in the X-ray light curves of both short ($\lesssim 2$ s) and long ($\gtrsim 2$ s) GRBs[26], and may be associated with the properties of the progenitors. While it is widely believed that short GRBs originate from compact binary merger[27], the progenitors of long GRBs are thought to be the explosion of (very) massive stars ($\gtrsim 10\ M_\odot$) emitting strong stellar winds[28,29]. This idea is strongly supported by GRB-SN associations[30,31], suggesting that Wolf-Rayet stars are the most likely progenitors of long-duration GRBs[32]. Additional supporting evidence are host galaxy studies[33] and relatively low metalicity[34]. The low metalicity implies the expected mass-loss rates to be smaller than those in typical Wolf-Rayet stars in our galaxy[35], $\dot{M} = 10^{-5} M_\odot \mathrm{yr}^{-1}$. This implies wind velocity of $v_w = 10^8\ \mathrm{cm\,s}^{-1}$ as a characteristics of a GRB progenitor. Indeed, multiple spectral components from the GRBs and SNe at the optical band are seen with speeds of $5 \times 10^7\ \mathrm{cm\,s}^{-1}$ and $3 \times 10^8\ \mathrm{cm\,s}^{-1}$[32]. These properties characterize the wind strength of the progenitor, therefore, affect the observed properties of the GRBs. As we show in this work, they may strongly affect the afterglow emission, and specifically the plateau emission.

Here we study in detail 13 GRBs with plateau phase seen in both X-ray and optical bands, selected from a sample of 222 GRBs with known redshifts and plateau phases defined in Ref. 7, see in methods subsection Sample Selection below. We consider a much simpler idea than previously discussed, which does not require any modification of the classical GRB fireball model. Rather, we simply look at a different region of the parameter space: a flow having an initial Lorentz factor of the order of a few tens, propagating into a wind (decaying density) ambient medium, with a typical density of up to two orders of magnitude below the expectation from a wind produced by a Wolf-Rayet star. We compute the physical parameters assuming synchrotron

emission from a power-law distribution of electrons accelerated at the forward shock. As we show, this model provides a natural explanation to the observed signals. We discuss the implication of the results on the properties of GRB progenitors and the resulting jets, and show how they provide a novel tool to infer the physical properties inside the jet.

## Results

### Sample selection and data analysis

We selected 13 GRBs based on the criteria defined in methods subsection Sample Selection below. In analyzing the optical and X-ray light curves (see Supplementary Method 1, Sample and data analysis), we identified two achromatic temporal breaks[6]: one during the transition from the plateau to the decaying light curve, which we interpret as transition from the coasting to the self-similar expansion (this break marks the end of the plateau and denoted by $T_a$); and a second, later break, which is identified as a jet-break, marked as $T_b$. In some bursts a second break could not be detected due to poor data sampling at very late times. The temporal slopes during the plateau phase, self-similar phase and after the jet break are marked as $\alpha_p$, $\alpha_{A1}$ and $\alpha_{A2}$ respectively and are given in Supplementary Tables 1–3.

### Theoretical regions

The optical and X-ray light curves do not necessarily follow the same power-law decay in either of the dynamical phases (plateau or self-similar). This can easily be understood in the framework of synchrotron emission from forward-shock accelerated electrons. The injected electrons assume a power-law distribution with power-law index $p$, namely $N_{el}(\gamma)d\gamma \propto \gamma^{-p}$ above a minimum value $\gamma_m$; below this value, one can assume the electrons to have a Maxwellian (or quasi-Maxwellian) energy distribution[9,36]. This assumption leads to a broken power-law spectra and light curves, whose shapes, in the relevant observed bands, depend on whether the peak frequency, $v_m$ (corresponding Lorentz factor $\gamma_m$) is above or below the cooling frequency, $v_c$ (corresponding Lorentz factor $\gamma_c$, for which the rate of energy lost by synchrotron emission is equal to the rate of energy lost by adiabatic cooling). For a given observed frequency (optical [typically the U-band]: $v_U$ or X-rays: $v_X$), different possibilities of the expected light curve and spectra exist. These possibilities are listed in Supplementary Table 7 and are displayed in Supplementary Fig. 17. The parameter space defined by the fast cooling regime ($v_m > v_c$) is split in three regions marked as A, B, C; while the parameter space for slow cooling regime ($v_m < v_c$) contains the regions marked as D, E, F, respectively. At low frequencies, one needs to consider synchrotron self-absorption, which can safely be neglected being below the optical frequency at all observed times (see details, methods subsection Theoretical model).

### Sample classification

After analyzing the data, we split the sample into 3 classes based on the X-ray and optical light curves during the plateau phase (corresponding to Supplementary Tables 1–3). These classes match very well the theoretical predictions. Class I is characterized by a flat X-ray light curve ($F_v \propto t_{obs.}^{0.0..-0.2}$), corresponding to regions C and F in Supplementary Table 7 and a decaying optical light curve ($F_v \propto t_{obs.}^{0.5..-0.7}$), region E in Supplementary Table 7. Theoretically, this class corresponds to GRBs having their cooling frequency between the optical and the X-ray bands, namely $v_U < v_c < v_X$. In class II, both the X-ray and the optical light curves are flat. Theoretically, this is expected for a low cooling frequency, $v_c < v_U < v_X$. In class III, both X-ray and optical light curves are decaying. This is expected when the cooling frequency is high, $v_U < v_X < v_c$. Interestingly, after excluding faint flares, all GRBs in our sample fall into one of these three classes. Furthermore, even tough the selection criteria for the different classes is based only on X-ray and optical light curves, there seems to be a connection between the energy of a burst, the duration of its plateau, and the its class. This is shown in Fig. 1.

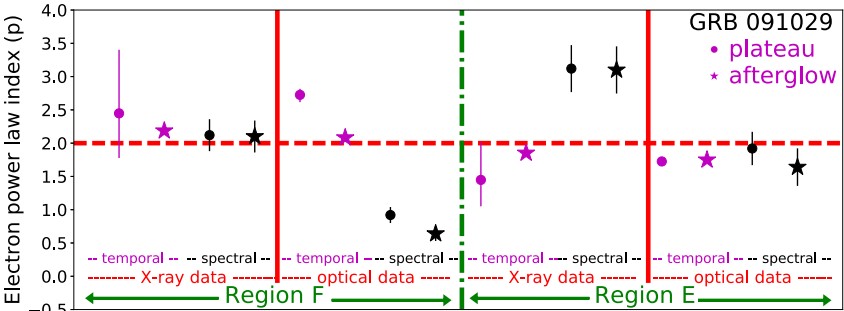

**Fig. 2 | An example demonstrating how we determine the relevant region for a given source as well as the electron power-law index (p) shown in Table 1.** For each burst (in the cases presented here, GRB 091029), we consider eight independent indicators, namely the temporal (purple) and spectral (black) slopes of both the X-ray and optical data during both the plateau (dot) and self-similar (star) phases. These eight independent measurements are inserted into the theoretical predictions given in Supplementary Table 7 to calculate the values of the electron power-law index, under the assumption that the data is in region F (left side) or E (right side). The obtained eight independent measurements (for each region, F and E), are then displayed side by side. The vertical green dash-dot line separates the analysis carried under the assumption that the emission is in region F (left) and E (right), the vertical red lines separates X-ray and optical data and the dashed horizontal line marks $p = 2$. The errors correspond to a significance of one sigma. For X-ray data, the independent calculations converge in region F to a single value of $p \simeq 2.15$ while the assumption that the outflow is in region E leads to a diverging result. Similarly, for optical data, the assumption that the emission is in region F leads to a divergence, while the assumption that it is in region E provides a consistent value of $p \simeq 1.85$. We therefore conclude that the X-ray emission of GRB 091029 is in region F, and the optical emission is in region E. Therefore, this burst is classified as being in class I. The source data necessary to reproduce this Figure are provided as a Source Data file.

## Closure relations and determination of the electron power-law indices

The theoretical model used herein imposes a relation between the spectral and the temporal slopes. These so-called closure relations are unique to each class and each observational band (optical and X-rays). Therefore, they can be used to assess the validity of our model. For each GRB, the X-ray spectral slopes ($\beta_p$ for the plateau phase, $\beta_{A1}$ for the self-similar phase, $\beta_{A2}$ for the spectral slope after the jet break) were obtained from the online Swift repository for the same time range as the temporal slopes, and optical spectral slopes were retrieved from the literature (the relevant references are given in Supplementary Table 1–3). We used the spectral and temporal slopes in each phase (plateau and self-similar) independently to further confront our theory to the data. Specifically, we checked that the closure relations relevant for regimes (C, E, F) for each band, each phase and each class are satisfied independently. The results are presented in Supplementary Figs. 15 and 16 for the plateau phase and the self-similar phase respectively. From these figures, it is clear that all the data––spectral and temporal, both in X-ray and optical - are consistent with the theoretical closure relations of the model.

Furthermore, from the closure relation, we deduced the power-law index $p$ of the accelerated electrons by using 8 independent measurements, namely the temporal and spectral indices of both the optical and X-ray data during both the plateau and self-similar phases. This was done for both regions E and F. As we demonstrate in Fig. 2, we find that the power-law index of the accelerated electrons does not change in between the dynamical phases. We find that for all GRBs in our sample, the power-law index, $p$ is in the range $1.8 \lesssim p \lesssim 2.5$. For those bursts whose X-ray light curve is identified as being in region E, the electron power-law index is narrowly clustered around $p \simeq 2$, while a larger spread is found for other GRBs. In 3 cases, namely GRBs 060614, 060729 and 110213A, we find that the values derived from the optical spectral slopes are inconsistent with the other measurements (see caption of Table 1), in which case we use 6 out of 8 independent measurements in determining the region and the power-law index. The results are summarized in Table 1.

We find that the optical data of 6 GRBs out of 13 in our sample (listed in Supplementary Table 2) are compatible with being in region F, implying that the cooling frequency at the end of the plateau phase is below the observed optical band, $\nu_c < \nu_U$. For another 4 GRBs out of 13 (listed in Supplementary Table 1), the optical light curve decays (corresponding to region E), while the X-ray light curve is flat, namely for these bursts $\nu_U < \nu_c < \nu_X$. For the remaining 3 GRBs out of 13 (listed in Supplementary Table 3) both the optical and X-ray light curves decay, and are therefore compatible with the cooling frequency being above the observed X-ray band $\nu_X < \nu_c$.

## The derived values of the physical parameters

We use here a simple theoretical model for the afterglow: the emission is produced by synchrotron radiation from electrons accelerated to a power law distribution at the forward shock, generated by the propagation of the ejecta into a wind medium, characterized by a decaying density: $\rho(r) = 5 \times 10^{11} A_\star r^{-2}$ g cm$^{-3}$. Here, the normalisation is obtained by assuming a wind mass-loss rate of $\dot{M} = 10^{-5} M_\odot \text{yr}^{-1}$ and a wind velocity of $v_w = 10^8$ cm s$^{-1}$ characteristics of a GRB progenitor (see, Supplementary Method 2, Theoretical model, Equation (9)). The excellent agreement between the data and the theory enables us to determine or constrain the parameters of the outflow and the wind, in particular the proportionality constant $A_\star$ of the wind density, the initial jet Lorentz factor, $\Gamma_i$, the fraction of energy in the electrons, $\epsilon_e$ and the magnetization, $\epsilon_B$. All relevant parameters used in the analysis are given in Table 2 (see, Supplementary Method 1c, Flux Ratio for details). For the 10 GRBs in Supplementary Tables 1 and 2, the X-ray flux and the transition time that marks the end of the plateau phase enable a direct deduction of $\epsilon_e$, while for the 3 GRBs in Supplementary Table 3, only a lower limit is available. For the 4 GRBs in Supplementary Table 1 we can directly infer the combined value of $A_\star \Gamma_i^4$. For a given value of $A_\star$, the value of $\epsilon_B$ is solely determined. Thus, an independent estimate of $\epsilon_B$ enables to break the degeneracy.

In Fig. 3 we use known limits ($10^{-5} \lesssim \epsilon_B \leq 0.1$)[37,38] to constrain the values of $A_\star$ and $\Gamma_i$. The lower values of $\epsilon_B$ are obtained from fitting the optical-to-X-ray light curves of bursts within the framework of the decaying afterglow model[37] as well as the analysis of bright LAT GRBs[39,40]. Such a low value of $\epsilon_B = 10^{-5}$ would not be obtained from the faint GRBs in class II due to the physical limitation on the Lorentz factor ($\Gamma > 1$). Therefore, we find that for GRBs in class II, $\epsilon_B$ cannot be smaller than $10^{-3}$. However, for the GRBs in classes I and III, such a restriction is not necessary, and $\epsilon_B$ can be as small as $10^{-5}$. Correspondingly, if indeed $\epsilon_B$ obtain such a low value, the density would be large. Assuming the value $\epsilon_B = 10^{-5}$, we show the micro physical parameters of GRBs in classes I and III in Fig. 3 in orange.

A tighter constraint on the minimum value of the coasting Lorentz factor $\Gamma_i$, thereby on the value of the magnetization of GRBs in class II can be put using the requirement that the prompt emission radius,

**Table 1 | Region and electron power-law index ($p$) in both X-ray and optical bands using both temporal and spectral indices**

| GRB name | Regions in X-ray band | | $p_X$ | | Regions in optical band | | $p_U$ | |
|---|---|---|---|---|---|---|---|---|
| | Temporal | Spectral | Temporal | Spectral | Temporal | Spectral | Temporal | Spectral |
| **Class I** | | | | | | | | |
| 080607 | F | F | ~2.5 | 2.0 | E | E | 2.0 | ~2.5 |
| 091029 | F | F | 2.3 | 2.0 | F(E) | E | 2.4(1.8) | 1.9 |
| 110213A | F | F | ~2.0 | 2.0 | E | E | 2.1 | 3.2 |
| 130831A | F | F/E | 2.0 | 1.8/~2.4 | E | — | 2.1 | — |
| **Class II** | | | | | | | | |
| 060605 | F | F | ~2.1 | 2.3 | F | F | ~2.2 | ~2.3 |
| 060614 | F | F | ~2.3 | 1.8 | F | F | 2.0 | 0.7 |
| 060729 | F | F | 2.0 | 2.0 | F | F | 2.3 | ~1.2 |
| 080310 | F | F | ~2.5 | 2.0 | F | F | 2.0 | 1.9 |
| 100418A | F | F | 2.2 | ~1.9 | F | F | ~1.8 | ~2.3 |
| 171205A | F | F/E | 2.0 | ~1.8/2.4 | F | F | ~1.8 | ~1.8 |
| **Class III** | | | | | | | | |
| 050319 | E | F/E | 2.1 | 2.1/3.2 | E | E | ~1.8–2.1 | 2.0 |
| 060714 | F/E | F/E | ~2.4/2.0 | 2.0/2.8 | F/E | E | 2.0/~1.8 | ~2.4 |
| 061121 | F/E | F/E | 2.5/~2.1 | 2.0/~3.2 | E | E | 2.0 | 2.3 |

In Column 1, GRB names are ordered by classes (I, II, III listed in Supplementary Tables 1–3 respectively). In columns 2–5, we use the temporal and spectral X-ray data to determine both the electron power-law index ($p_X$) and the region (E or F) characteristics of the emission, see Supplementary Fig. 17. The optical data is used in a similar way in columns 6-9. For GRBs 060614, 060729 and 110213A, the power-law indices obtained using X-ray and optical temporal indices as well as the X-ray spectral index are all consistent with each other, while the values derived using the optical spectral data deviate. Since no errors are given in the literature for GRBs 060614 and 060729 (see Supplementary Table 2), we cannot estimate the reliability of the optical spectral indices for these bursts, and we therefore accept the power law obtained using 6 out of 8 independent measurements. For GRB 110213A, there is no available spectral index in the optical band during the plateau phase, and the second peak in the optical band after the plateau phase (see, Supplementary Method 1b, Optical data and fitting process) might be affecting the spectral index during the afterglow phase (see Supplementary Table 1). In all three bursts which we categorize as being in class III, there seem to be a discrepancy between the X-ray spectral and temporal data: while the temporal data clearly indicates the X-ray to be in region E the spectral data favours region F. However, this is because, in all three bursts there is a break in the X-ray light curve during the plateau phase, or an early flare. These may indicate a change in region, and may affect the spectral measurement.

**Table 2 | Some key parameters of the 13 GRBs in our sample**

| GRB | $z$ | $\log(E_{iso}/\mathrm{erg})$ | $T_{a,X}$ ($10^3$ s) | $\nu F_\nu(X)$ ($10^{-12}$ erg cm$^{-2}$ s$^{-1}$) | $\nu F_\nu(U)$ ($10^{-12}$ erg cm$^{-2}$ s$^{-1}$) | $T_{ref,U}$ (s) | $\nu F_\nu(X)$ ($10^{-12}$ erg cm$^{-2}$ s$^{-1}$) | $\nu F_\nu(U)$ ($10^{-12}$ erg cm$^{-2}$ s$^{-1}$) |
|---|---|---|---|---|---|---|---|---|
| **Class I** | | | | | | | | |
| 080607 | 3.036 | $53.27^{\pm0.02}$ | $2.23^{+0.32}_{-0.27}$ | $56^{\pm12.6}$ | $0.13^{\pm0.01}$ | 1010 | $82.7^{\pm18.5}$ | $0.18^{\pm0.01}$ |
| 091029 | 2.752 | $52.31^{\pm0.16}$ | $14.1^{+1.59}_{-3.37}$ | $1.34^{\pm0.23}$ | $0.117^{\pm0.029}$ | 1170 | $2.8^{\pm0.6}$ | $0.3^{\pm0.1}$ |
| 110213A | 1.46 | $52.45^{\pm0.06}$ | $1.35^{+0.14}_{-0.18}$ | $350^{\pm77}$ | $8.45^{\pm0.33}$ | 1130 | $218^{\pm48}$ | $7.7^{\pm0.4}$ |
| 130831A | 0.4791 | $51.57^{\pm0.01}$ | $0.75^{+0.08}_{-0.07}$ | $259^{\pm56}$ | $46.2^{\pm2.8}$ | 732 | $259^{\pm56}$ | $46.2^{\pm2.8}$ |
| **Class II** | | | | | | | | |
| 060605 | 3.78 | $52.00^{\pm0.15}$ | $4.88^{+1.24}_{-1.85}$ | $2.0^{\pm0.45}$ | $1.12^{\pm0.114}$ | 534 | $16.3^{\pm3.3}$ | $9.51^{\pm0.42}$ |
| 060614 | 0.125 | $50.87^{\pm0.01}$ | $34.1^{+2.28}_{-2.64}$ | $3.28^{\pm0.63}$ | $0.626^{\pm0.143}$ | 4838 | $2.1^{\pm0.5}$ | $0.50^{\pm0.12}$ |
| 060729 | 0.54 | $51.25^{\pm0.04}$ | $38.1^{+3.35}_{-2.11}$ | $6.35^{\pm1.43}$ | $3.34^{\pm0.75}$ | 1160 | $10.6^{\pm2.4}$ | $4.13^{\pm0.94}$ |
| 080310 | 2.42 | $52.67^{\pm0.09}$ | $10.9^{+0.94}_{-0.88}$ | $3.41^{\pm0.76}$ | $0.258^{\pm0.059}$ | 1505 | $4.9^{\pm0.9}$ | $1.41^{\pm0.32}$ |
| 100418A | 0.6235 | $50.57^{\pm0.08}$ | $79.3^{+20.6}_{-12.8}$ | $0.86^{\pm0.22}$ | $0.258^{\pm0.005}$ | 1000 | $0.15^{\pm0.04}$ | $0.06^{\pm0.01}$ |
| 171205A | 0.0368 | $49.03^{\pm0.04}$ | $91.0^{+8.60}_{-8.45}$ | $1.02^{\pm0.24}$ | $0.568^{\pm0.068}$ | 10834 | $0.61^{\pm0.16}$ | $1.44^{\pm0.09}$ |
| **Class III** | | | | | | | | |
| 050319 | 3.24 | $52.48^{\pm0.13}$ | $32.0^{+4.36}_{-4.27}$ | $1.24^{\pm0.28}$ | $0.144^{\pm0.026}$ | 1120 | $5.02^{\pm1.13}$ | $1.20^{\pm0.17}$ |
| 060714 | 2.711 | $52.64^{\pm0.10}$ | $4.54^{+1.38}_{-1.01}$ | $13.3^{\pm2.92}$ | $0.180^{\pm0.053}$ | 1069 | $13.6^{\pm3.07}$ | $0.31^{\pm0.08}$ |
| 061121 | 1.314 | $52.58^{\pm0.01}$ | $9.11^{+0.53}_{-0.59}$ | $61.5^{\pm12.3}$ | $0.916^{\pm0.250}$ | 1173 | $67^{\pm15}$ | $1.66^{\pm0.40}$ |

Columns 1–9 are the GRB names (ordered by classes I, II, III listed in Supplementary Tables 1–3 respectively), redshift, isotropic equivalent energy in log scale, time at the end of the X-ray plateau phase, $\nu F_\nu$ X-ray flux and $\nu F_\nu$ optical flux at $T_{a,X}$, a reference time in optical band at around 1000 s, $\nu F_\nu$ X-ray flux and $\nu F_\nu$ optical flux at $T_{ref,U}$ respectively. The errors correspond to a significance of one sigma. See, Supplementary Method 1c, Flux Ratio for the definition of each parameters.

$R_E = 2\Gamma_i^2 c\Delta t_{min}$ be above the photospheric radius, $R_{ph} = L_{iso}\sigma_T/(8\pi m_p c^3 \Gamma_i^3)$[41]. Here, $\Delta t_{min}$ is the minimum observed variability timescale during the prompt phase, $L_{iso}$ is the isotropic luminosity, $\sigma_T$ is the Thomson cross section, $m_p$ is proton mass and $c$ is the speed of light. For typical GRB parameters (including sub-second variability, $\Delta t_{min} = 0.1$ s and isotropic luminosity, $L_{iso} = 10^{50.5}$ erg/s), the requirement $R_E \geq R_{ph}$ results in a minimum Lorentz factor $\Gamma_i \gtrsim 30$[41,42]. A few GRBs in our sample, mainly in class II have an estimated Lorentz factor

lower than this limiting value. However, all these GRBs are low luminosity GRBs, implying relatively low signal-to-noise ratio ($S/N \leq 10$) during the prompt phase[43]. Only in one case (GRB 060729) a reliable variability is measured, giving $\Delta t_{min} = 4.99$ s[43], although the general trend of low luminosity GRBs having $\Delta t_{min} > 1.0$ s is clearly observed[43,44]. Using the *Swift*-BAT light curves of all other low $\Gamma_i$ GRBs in our sample we estimate that the observed variability of these GRBs is much longer, and we choose as a conservative estimate $\Delta t \geq \Delta t_{min} = 5$ s. Most importantly,

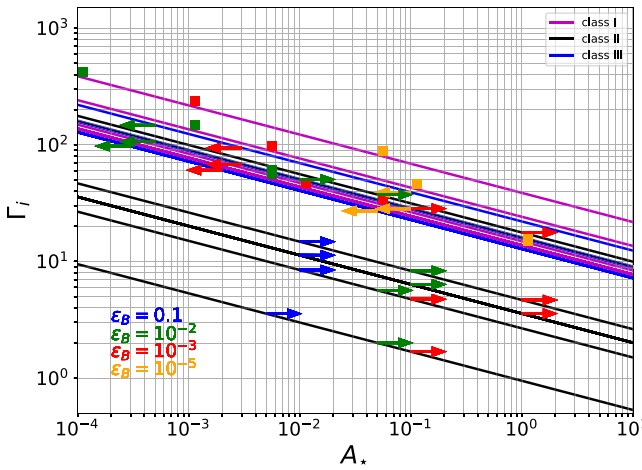

**Fig. 3 | The initial GRB jet Lorentz factor, $\Gamma_i$ and the ambient density, $A_\star$.** They are marked by lines, each corresponds to a different GRB. From top to bottom: GRBs 080607, 110213A, 060714, 080310, 061121, 060605, 130831A, 091029, 050319, 060729, 060614, 100418A, and 171205A. Purple, black and blue lines represent the GRBs in classes I, II and III respectively. For a given $A_\star$, we determine the value of $\Gamma_i$ by using Supplementary Equation (20) and knowing the burst energy and transition time $T_{a,x}$. This gives the lines. In order to further constrain the values of these parameters, we assume knowledge of magnetization, $\epsilon_B$, and use Supplementary Equation (21) to deduce direct values of $A_\star$ and $\Gamma_i$ (squares) for class I. For classes II and III, we instead use Supplementary Equations (24) and (25) to compute the lower (upper) and upper (lower) limits of $A_\star$ ($\Gamma_i$) respectively. These limites are represented by arrows. In all classes, the constraint put by the magnetization ($\epsilon_B$) is inversely proportional to the ambient density. We mark on the plot the values obtained for $\epsilon_B = 0.1, 10^{-2}, 10^{-3}$ and $10^{-5}$ which are associated to the blue, green, red and orange colors, respectively. The lowest value of Lorentz factor (4, blue arrow) is obtained for GRB 171205A. Due to its low luminosity, $L_{iso} = 5.6 \times 10^{46}$ erg/s, the prompt emission radius is above the photospheric radius, as shown in Fig. 4. In addition, this GRB is found to exhibit a black-body emission with a low temperature in the X-ray spectra, later on this component cooling into the UV and optical range over time[70,71]. The source data to reproduce this figure are provided as a Source Data file.

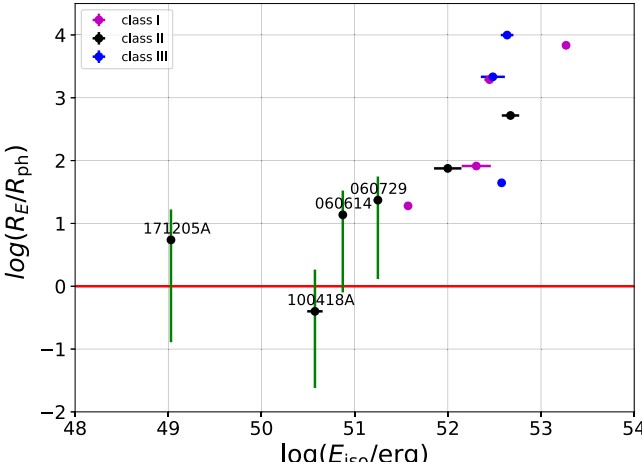

**Fig. 4 | Ratio of the prompt emission radius, $R_E$ and the photospheric radius, $R_{ph}$ versus total isotropic energy $E_{iso}$.** Purple, black and blue points represent GRBs in the three different classes I, II, III respectively (see Supplementary Tables 1–3). The errors correspond to a significance of one sigma. The low luminous GRBs with lowest Lorentz factor (GRBs 060614, 060729, 100418A and 171205A) are marked by their names. For these GRBs we consider $\epsilon_B = 0.1$, leading to typical initial jet Lorentz factor $\Gamma_i \approx 10$ and ratio $R_E/R_{ph} \sim 5 - 25$, but for GRB 100418A, which is marginally consistent with $R_E/R_{ph} \sim 1$. For all other GRBs, we assume $\epsilon_B = 10^{-3}$ when making this plot, and a variability time $\Delta t_{min}$ taken from Refs. 43, 44, except for GRB 060714, where $\Delta t = 5$ s is assumed based on data from the *Swift*-BAT light-curve. We point out that a higher value of $\epsilon_B$ increases this ratio. The vertical green lines associated to each GRB illustrate the possible ratios of $R_E/R_{ph}$ obtained for magnetization in the range $10^{-3} \leq \epsilon_B \leq 0.7$. The horizontal red line indicates $R_E/R_{ph} = 1$. It shows that in all cases, sufficiently high value of $\epsilon_B$ within the examined range leads to $R_E > R_{ph}$. The source data to reproduce this figure are provided as a Source Data file.

given the degeneracy between $\Gamma_i - A_\star$, for the GRBs with the lowest Lorentz factor, namely GRBs 060614, 060729, 100418A and GRB 171205A the required criteria is achieved for value of $\epsilon_B \geq 0.1$, which enforces (relatively) high $\Gamma_i$ and low $A_\star$ (the left hand side in Fig. 3). We therefore mark the value $\epsilon_B = 0.1$ by blue arrows only for these GRBs in Fig. 3. The derived Lorentz factors for these bursts are $\approx 10$, while the density is characterized by $A_\star = 10^{-2}$. The ratio of prompt emission radius to photospheric radius of all GRBs in our sample are plotted in Fig. 4, demonstrating that despite the low values of the Lorentz factors we obtain, the prompt emission radius is always above the photospheric radius.

For the 6 GRBs in Supplementary Table 2, only a lower limit on the value of $\epsilon_B$ can be deduced. This is still highly valuable, as physically $\epsilon_B < 1.0$. For the 3 GRBs in Supplementary Table 3, only an upper limit on $\epsilon_B$ is obtained. We use these limits to constrain the combination of $A_\star \Gamma_i^4$. They are presented in Table 3 and Fig. 3. In the figure, the values of $\Gamma_i$ and $A_\star$ are marked by lines, each correspond to a different GRB. These are, from top to bottom: GRBs 080607, 110213A, 060714, 080310, 061121, 060605, 130831A, 091029, 050319, 060729, 060614, 100418A and 171205A. While directly deduced values are marked by squares, upper and lower values are marked by arrows.

## Discussion

A histogram of the initial Lorentz factor $\Gamma_i$ for the 13 GRBs in our sample is shown in Fig. 5. The average value of the Lorentz factor deduced is $\langle \Gamma_i \rangle \approx 51$ (the median is 32), although the range span is between $2 \lesssim \Gamma_i \leq 218$. These values may initially seem at odds with the typical

values discussed in the literature, of $\Gamma_i \gtrsim 100$ of GRB jets. However, a closer look reveals that in fact there is no contradiction.

There are several ways of inferring the value of the Lorentz factor in GRB jets. The most widely used method is the opacity argument[1,2], which is commonly used in deducing that observed GeV photons by *Fermi*-Large Area Telescope (LAT) must originate from a region expanding with a Lorentz factor $\Gamma_i \gtrsim 100's$. This argument, though, is only valid when > MeV photons are observed. A previous search[45] of GRBs observed from 2008 until May 2016 by *Fermi*-LAT that appeared in the $2^{nd}$ catalogue[46] and are (1) fitted with a broken power-law, (2) have the Test Statistic, $TS > 64$ and (3) have known redshift (in total 13 GRBs) demonstrates that although 3 GRBs out of the thirteen show evidence for a shallow decay phase in the LAT data, only one (the hard-short GRB 090510) show any evidence for a decaying plateau in the *Swift*-XRT data. For this specific burst, the shallowest decay segment in its X-ray afterglow light-curve can only marginally be associated to a plateau, having an X-ray slope of $-0.69^{+0.05}_{-0.06}$, to be compared to the $-0.7$ limit used in this study. These arguments are consistent with earlier findings by Ref. 47 for 23 GRBs triggered by *Swift*-BAT and subsequently detected by *Fermi*-LAT[46].

The second method relies on identifying an early optical flash and interpreting it as originating from the reverse shock[3,48,49]. Since the reverse shock exists during the transition from the coasting to the decaying (self-similar) phase, identifying its emission constrains the transition time, from which, assuming the energy and ambient density are known, the initial Lorentz factor can be deduced. However, a clear signature of a reverse shock emission is nearly never identified[50,51] as opposed to flares common to both the X-ray and optical data[52–55] and does not exist in any of the bursts in our sample.

When a strong thermal component exists during the prompt phase, it is possible to use it to infer the Lorentz factor at the initial

**Table 3 | Model parameters of the outflow and wind**

| Class I | $\frac{\epsilon_e}{10^{-2}}$ | $\epsilon_B = 10^{-2}$ | | $\epsilon_B = 10^{-3}$ | | $\epsilon_B = 10^{-5}$ | | Ratio $\nu F_\nu(X)/\nu F_\nu(U)$ |
|---|---|---|---|---|---|---|---|---|
| | | $\Gamma_i$ | $\frac{A_\star}{10^{-2}}$ | $\Gamma_i$ | $\frac{A_\star}{10^{-2}}$ | $\Gamma_i$ | $A_\star$ | |
| 080607 | 1.2 | 387 | 0.01 | 218 | 0.1 | 122 | $10^{-2}$ | $314 \pm 74$ |
| 091029 | 1.4 | 52 | 0.5 | 44 | 1 | 14 | 1.0 | $4.5 \pm 1.4$ |
| 110213A | 8.3 | 136 | 0.1 | 91 | 0.5 | 43 | 0.1 | $45.4 \pm 2.2$ |
| 130831A | 2.7 | 56 | 0.5 | 32 | 5 | 15 | 1.0 | $5.6 \pm 1.3$ |

| Class II | $\frac{\epsilon_e}{10^{-2}}$ | $\epsilon_B = 10^{-2}$ | | $\epsilon_B = 10^{-3}$ | | Ratio $\nu F_\nu(X)/\nu F_\nu(U)$ |
|---|---|---|---|---|---|---|
| | | $\Gamma_i$ | $\frac{A_\star}{10^{-2}}$ | $\Gamma_i$ | $A_\star$ | |
| 060605 | 2.5 | <51 | >1 | < 28 | > 0.1 | $0.21 \pm 0.05$ |
| 060614 | 0.49 | <6 | >10 | < 4 | > 1.0 | $6.6 \pm 2.0$ |
| 060729 | 9.2 | <8 | >10 | < 5 | > 1.0 | $1.5 \pm 0.5$ |
| 080310 | 1 | <37 | >5 | < 32 | > 0.1 | $2.4 \pm 0.8$ |
| 100418A | 17 | <6 | >5 | < 5 | > 0.1 | $13.7 \pm 4.6$ |
| 171205A | 2.3 | <2 | >5 | < 1.7 | > 0.1 | $0.7 \pm 0.2$ |

| Class III | $\epsilon_B = 10^{-2}$ | | | $\epsilon_B = 10^{-3}$ | | | $\epsilon_B = 10^{-5}$ | | | Ratio $\nu F_\nu(X)/\nu F_\nu(U)$ |
|---|---|---|---|---|---|---|---|---|---|---|
| | $\frac{\epsilon_e}{10^{-1}}$ | $\Gamma_i$ | $\frac{A_\star}{10^{-4}}$ | $\frac{\epsilon_e}{10^{-3}}$ | $\Gamma_i$ | $\frac{A_\star}{10^{-3}}$ | $\frac{\epsilon_e}{10^{-7}}$ | $\Gamma_i$ | $\frac{A_\star}{10^{-2}}$ | |
| 050319 | >31 | >97 | <3 | >16 | >61 | <2 | >21 | >27 | <5 | $1.03 \pm 0.27$ |
| 060714 | >0.8 | >147 | <5 | >0.55 | >94 | <3 | >0.33 | >39 | <10 | $43.6 \pm 14.8$ |
| 061121 | >2.8 | >106 | <5 | >1.9 | >70 | <3 | >1.1 | >28 | <10 | $37.1 \pm 11.6$ |

In Column 1, GRB names are ordered by classes (I, II, III listed in Supplementary Tables 1–3 respectively). $\epsilon_e$ is the fraction of energy in the electrons, $\Gamma_i$ is the initial jet Lorentz factor, $A_\star$ is the wind density. Direct value of $\epsilon_e$ is computed by using the information in the X-ray data for the GRBs listed in class I and II respectively. The values obtained (using the end of plateau time and X-ray flux in Supplementary Equation (19)) are surprisingly close to the fiducial values, $\epsilon_e \simeq 10^{-1}$ often obtained by fitting late time afterglow data[61]. In addition, direct values of $A_\star$ and $\Gamma_i$ are obtained by assuming that the fraction of energy in the magnetic field is $\epsilon_B = 10^{-2}, 10^{-3}, 10^{-5}$ for the GRBs listed in class I. Moreover, an (external) upper limit on $\epsilon_B$, (e.g., $\epsilon_B \leq 10^{-2}$) is used to compute an upper limit on $\Gamma_i$ and lower limit on $A_\star$ for the GRBs listed in class II. Vice-versa, an external knowledge on lower limit (e.g., $\epsilon_B \geq 10^{-5}$) can be used to compute a lower limit on the value of $\Gamma_i$ and an upper limit on $A_\star$ as well as a lower limit on $\epsilon_e$ for the GRBs listed in class III (see, Supplementary Method 2, Theoretical model). The ratio $[1/(\nu F_\nu(U)/\nu F_\nu(X))]$ is such that $\nu F_\nu(U)$ is calculated at $T_{\rm ref., U}$ and $\nu F_\nu(X)$ is calculated at $T_{a,X}$ (see Table 2). The errors correspond to a significance of one sigma. The ratios are consistent with the theoretical predictions in all three different classes. For the low luminous GRBs with the lowest Lorentz factor (GRBs 060614, 060729, 100418A and 171205A) in class II, when considering $\epsilon_B = 0.1$, the obtained values are $\Gamma_i = 11, 15, 9, 4$ and $A_\star = 10^{-2}, 10^{-2}, 10^{-2}, 5 \times 10^{-3}$, respectively.

phase of the expansion[56]. We therefore searched (i) all bursts with known strong thermal component as appeared in Refs. [57–59]. None of those showed any evidence for an X-ray plateau. (ii) Similarly, none of the bursts in our sample show any evidence for a thermal emission.

To conclude, we find that in all cases where there is any evidence for an initial Lorentz factor $\Gamma_i \gtrsim$ a few hundreds, no X-ray plateau exists, and vice versa: for all bursts that show a plateau, no significant indication for high Lorentz factor exist, neither at high energy, thermal or optical photons. We further emphasis that GRBs with plateau phase which have very low Lorentz factor (namely, GRBs in class II) lack any evidence of MeV emission, implying that the opacity argument cannot be used at all in these bursts.

One may argue that such a difference in the Lorentz factor cannot only manifest itself in the afterglow phase, but should be manifested in the prompt emission spectra as well. To further test this hypothesis, we therefore compared the peak energy of the 13 bursts in our sample to a reference sample of selected 12 GRB without plateau phase, as presented in table 6 of Ref. [51]. In that work, the authors estimated the Lorentz factor of these bursts using X-ray onset bump or early peak in the optical data and found that the Lorentz factor is of the order of few hundreds in all cases. We therefore concluded that this is a good reference for comparing the distribution of observed peak energies, $E_{\rm pk}$. In order to ensure consistency, we did not use the values of $E_{\rm pk}$ as given in Ref. [51] (measured from different instruments e.g. Konus-WIND). Rather, we calculated $E_{\rm pk}$ directly from the *Swift*-BAT data. This ensures that there are no biases between the samples. In the calculation, we used the correlation between the peak energy and spectral index derived from fitting a single power-law to a large *Fermi*-GBM and *Swift*-BAT data as parameterized by Ref. [60] as such a method is commonly used in the literature[61,62]. The method gives consistent results ($E_{\rm pk}$) with the deduced value from the *Fermi*-GBM data. In Fig. 6, we compare the peak energy distribution of these two samples. A clear

separation is found: those GRBs which have a higher Lorentz factor indeed have a consistently higher $E_{\rm pk}$ than the GRBs in our sample. In addition to the clear differences between the peak energy distribution of plateau and without plateau, we also found clear differences between the high energy spectral indices (presented in the *Fermi*-GBM GRB catalog[63]) of bursts in these samples.

Furthermore, it is known that short GRBs have, on the average, higher $E_{\rm pk}$ than long GRBs[59], while it was argued by Ref. [7] that 43/222 short GRBs do show a plateau. However, we point out that the criteria used by Ref. [7] for a plateau, namely a break in the X-ray light curve, is much less restrictive than the one used here (namely, X-ray temporal index >−0.7). Using the criteria in this work, none of the short GRB light curves considered by Ref. [7] would be classified as having a flat X-ray light curve (considered as class II, with low Lorentz factor of $\gtrsim$ few), or was observed in the MeV range by *Fermi*-GBM[63]. Therefore, the compactness argument does not apply for these bursts.

The results we have here, therefore, complement and extend the known range of Lorentz factors in GRBs. The values of $A_\star$ we find are typically up to 2 orders of magnitude lower than the fiducial value of $A_\star = 1$ (pending on the exact value of the magnetization parameter, $\epsilon_B$; see Fig. 3). We therefore conclude that the expansion occurs into a low-density wind, having density which may be somewhat lower than the expectation from a Wolf-Rayet star ($A_\star \simeq 0.5 - 1.0$)[64,65]. This result therefore implies that either Wolf-Rayet stars are not the progenitors of GRBs with plateau, or that the properties of the wind ejected by the star prior to its final collapse are different than in earlier stages of its life.

Indeed, we cannot consider this as an evidence against Wolf-Rayet progenitor stars, as very little is known about the final stages of the evolution of the most massive stars (luminous blue variables and Wolf-Rayet stars), of which some lead to an evolutionary channel which end up as GRBs. Rapid evolutionary stages of such stars are expected during the last 10's of centuries of their life, which will have profound

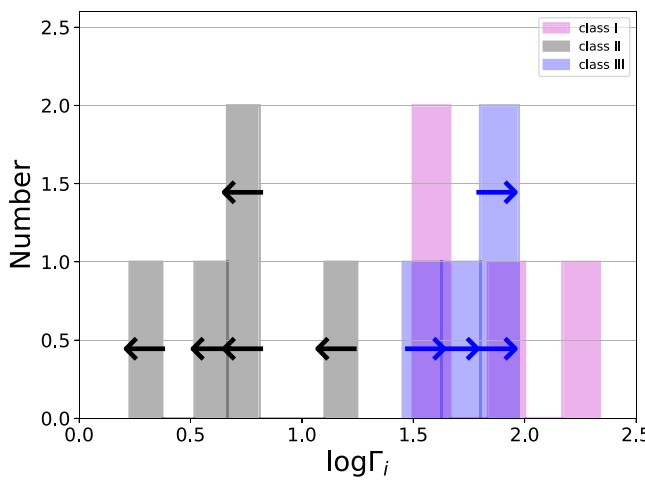

**Fig. 5 | Histogram of the initial jet Lorentz factors for the 13 GRBs in our sample.** These values are obtained by assuming that the fraction of energy in the magnetic field is $\epsilon_B = 10^{-3}$. The purple bars represent values deduced directly from the data (class I), the black bars are upper limits (class II) and the blue bars are lower limits (class III). Upper and lower limits are also marked by arrows. The average value of the initial GRB jet Lorentz factor is $\langle \Gamma_i \rangle \approx 51$ (median is 32), although the range span is between $1.7 \lesssim \Gamma_i \lesssim 218$ (see Table 3). We point out that GRB 080607 which has the highest value of $\Gamma_i$ has a large gap in its X-ray LC between the plateau and self-similar phases. Furthermore, GRB 171205A which has the lowest value of $\Gamma_i$ is associated with SN 2017iuk, therefore, both the optical plateau and self-similar slopes of this burst are effected by the SN bump (see, Supplementary Method 1b, Optical data and fitting process for further discussion). The source data to reproduce this figure are provided as a Source Data file.

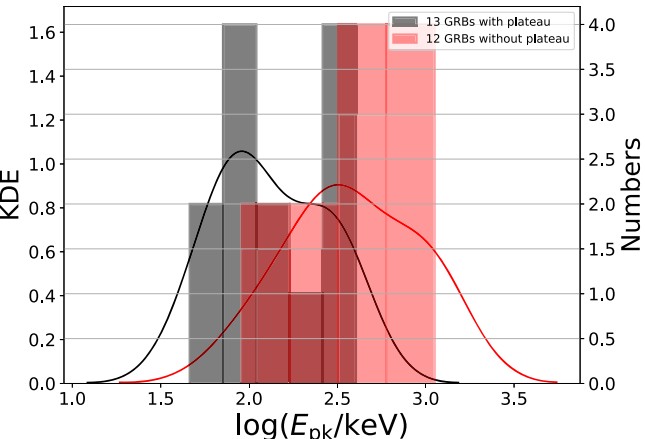

**Fig. 6 | Distributions of peak energy, $E_{pk}$.** The black bars represent the 13 GRBs with plateau phases in our sample, the red bars represent 12 GRBs without plateau phases presented in Table 6 of Ref. 51. Note that GRBs 050319, 061021 as well as early GRBs observed before the launch of *Swift*-BAT in Ref. 51 were discarded. In the panel the right-hand ordinate is the number of burst in each histogram bin and the left-hand ordinate is the value of the kernel density estimation (KDE), which is shown by the black and red curves for each sample respectively. Clearly, plateau bursts have a lower peak energy than other bursts. This result is consistent with the idea that jets in these bursts have a lower initial jet Lorentz factor. We also performed a Kolmogorov-Smirnov test (KS test: $D = 0.60$ and $p = 0.014$) which can clearly show if these two samples originate from the same population. The KS test result shows that there is only 1.4% chance for these samples to have originated from the same population and they differ with 60%. We view this as another hint towards understanding the difference between GRBs that do show a plateau and those do not. The source data to reproduce this figure are provided as a Source Data file.

affect on the circumstellar wind profiles. Instabilities will cause elevation of the outer envelope potentially leading to occasional giant eruption events, with major mass ejections in several consecutive periods. These mass ejections lead to circumstellar nebulae and wind blown bubbles[66,67]. Observations of galactic Wolf-Rayet stars indicate shell structures and nebulae at 1-10 pc scales, and in some cases, reveals the existence of low density cavities within these nebulae[67]. We thus view one of the merits of this work as providing further information that could potentially help understanding the nature of these objects.

Clearly, the fact that a substantial fraction of GRBs have a Lorentz factor of tens rather than hundreds bridges an important observational gap. Other astronomical objects known to have jets such as X-ray binaries or active-galactic nuclei (AGNs) have mildly relativistic jets, with $\Gamma_i \lesssim 20$, while earlier estimates of the initial Lorentz factor in GRB jets are in the hundreds. Our result, therefore implies that the range of initial jet velocities that exist in nature does not have a 'gap' in the range $\Gamma_i$ of tens, but is rather continuous from mildly relativistic to $\lesssim 1000$. This is shown by the histogram presented in Fig. 7.

Here, we consider a simple model in explaining the X-ray plateau in GRB afterglows, which does not require any modification of the classical GRB fireball model. Rather, we simply look at a different region of the parameter space: we consider an outflow having an initial Lorentz factor of the order of few tens, propagating into a wind environment, with a typical density of up to two orders of magnitude below the expectation from a wind produced by a Wolf-Rayet star. We follow a similar idea that was proposed by Ref. 68, but did not gain popularity, as (i) the deduced values of the Lorentz factor are lower than the fiducial values, $\Gamma_i \gtrsim 100$; and (ii) it was mistakenly claimed that this model can only account for achromatic afterglow, and can therefore explain only a sub-sample of the GRB population[25]. As we showed here, (i) there is no contradiction in the deduced value of the Lorentz factor, and (ii) the claim for an achromatic afterglow break is incorrect, as the optical and X-ray bands are not necessarily in the same regimes.

In our work, we considerably extended this simple idea theoretically and thoroughly confronted it to observations. We show that whenever there is enough data to perform a fit in both X-ray and optical bands, the break time in between these two bands is compatible and data in both bands can be interpreted within the single theoretical model presented here. We further carried out a more careful analysis on a much larger data set, allowing for a larger freedom (with more than a single break) and removal of flares on both the X-ray and optical data. We extended the theory to include all possible regimes. We showed that all observed light-curves can be explained by at least one of these regimes. We then showed how the confrontation of our model to the data can be used to infer the values of the density, Lorentz factor, magnetization and fraction of energy carried by the electrons. Moreover, our model provides several testable predictions about bursts with long plateau. Such bursts (i) are not expected to show high energy ($\gtrsim$ GeV) emission; (ii) are not expected to show strong thermal component; and (iii) the typical variability time during the prompt phase is expected to be long, of the order of few seconds. Exact constraints can be put on a case-by-case basis, using the equations we provide below (see, methods subsection Theoretical model).

While the idea presented in this work is very simple, clearly it has very far reaching consequences: [a] On the nature of long GRB progenitors, which can either (i) not be a Wolf-Rayet star, or (ii) imply that the properties of the wind ejected by these stars prior to their final explosion is very different than the properties of the wind ejected at earlier times. [b] On our understanding of the nature of the explosion itself, which produce a much wider range of initial jet Lorentz factor, which in many cases are in the range of tens and in others are in the hundreds.

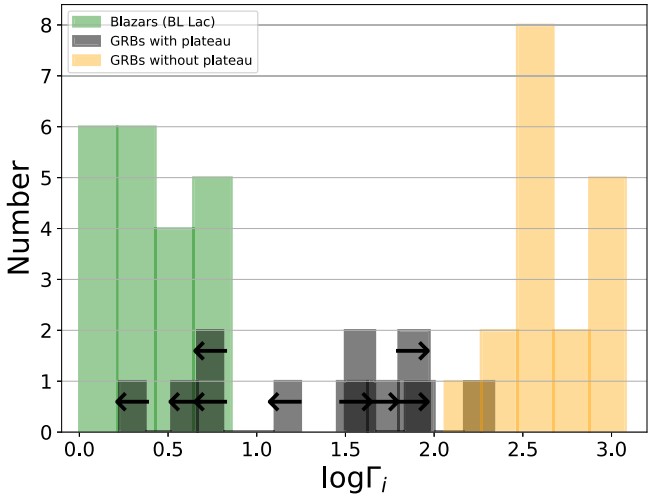

**Fig. 7 | Histogram of the initial jet Lorentz factors.** Black represents the GRBs in our sample, as they appear in Fig. 5. Yellow represents GRBs observed by the Fermi-LAT instrument, taken from Ref. 3, whose jet Lorentz factor is deduced from the opacity argument (GRBs 090926A, 090902B, 090510, 080916C; see Ref. 3, figure 11) and other GRBs that do not show a plateau phase (except GRBs 050319 and 061021), taken from Ref. 51 (table 6 therein). Green represents the inferred lower limit to the Lorentz factor of BL Lac objects (Blazars), taken from Ref. 72 (see their table 2A). The values of the Lorentz factor we found, of few tens, fills the gap that previously existed in the range of initial jet velocities. The source data to reproduce this figure are provided as a Source Data file.

## Methods

### Sample selection

To study the properties of the plateau, we use the sample of 222 GRBs with known redshifts and plateau phases defined in Ref. 7. These GRBs were detected by the Neil Gehrels *Swift* Observatory from January 2005 until August 2019. They represent 56% of all GRBs with known redshifts observed by the *Swift* satellite in this period.

In order to make our analysis as reliable as possible, we limit the bursts used to the ones having the best quality observations. Therefore, we added three criteria to the ones used by Ref. 7. We require (i) a long lasting (from $10^2$ to $10^5$ s) plateau phase with a temporal X-ray slope larger than −0.7, followed by a power-law decay phase at later times (interpreted here as the self-similar phase). (ii) Sufficient number of data points (≳5) during the plateau and self-similar phases to enable the fits to give well constrained parameter (see, Supplementary Method 1a, X-ray data and fitting process). For the analysis to be valid, we excluded all X-ray flares from the analyzed data. We found 130 GRBs matching these two criteria. (iii) Finally, we require an optical counterpart at around the same time as the X-ray data. We searched the optical catalogue of Ref. 69 and found that 24 GRBs in our sample have an optical counterpart. Out of these, we had full access to the optical data of 13 GRBs, which are listed in Supplementary Tables 1–3. The analysis of X-ray and optical light curves of these 13 GRBs is detailed in Supplementary Method 1, Sample and data analysis.

### Theoretical model

The key to understanding the observations in the framework of our model is the realization that the end of the plateau corresponds to the transition from a coasting phase (steady state in which all the energy has been converted to kinetic energy) to a self-similar expansion phase (decaying phase in which the kinetic energy is being converted back to radiation by the shocks) of the expanding plasma. In this model, the emission originates entirely from ambient electrons collected and heated by the forward shock wave, propagating at relativistic speeds inside a wind (decaying density) ambient medium. As we show (see, Supplementary Method 2, Theoretical model) this assumption about

the decay of the ambient density is crucial in explaining the observations. During the transition from the coasting phase to the decaying phase a reverse shock crosses the expanding plasma. However, the contribution from electrons heated by the reverse shock is suppressed due to (i) the declining ambient density, which implies that the ratio of plasma density to ambient density remains constant (under the assumption of a conical expansion), and (ii) its slower speed, which translates into less energetic electrons that emit at much lower frequencies than forward shock heated electrons, implying that the contribution to the optical and X-ray bands is negligible. A detailed derivation of the theoretical model is provided in Supplementary Method 2, Theoretical model.

## Data availability

The data used in this paper are publicly available via https://www.swift.ac.uk/xrt_curves/ in X-ray band and references listed in Supplementary Method 1b, Optical data and fitting process for optical band. The processed data that support the findings of this study are available from the corresponding author upon reasonable request. The X-ray and the optical light curves of 13 GRBs with the overlaid fit parameters are presented in the Supplementary Information File as Supplementary Figures. The source data for all figures in the main manuscript and for the Supplementary Figs. 15 and 16 are provided with this paper. The authors declare that all other data supporting the findings of this study are available within the paper and its supplementary information files. Source data are provided with this paper.

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

## Acknowledgements

We thank Dr. Damien Bégué, Dr. Mukesh Vyas and Dr. Filip Samuelsson for their comments throughout the development of this work. This work made use of data supplied by the UK Swift Science Data Centre at the University of Leicester. H.D.-B. and A.P. is supported by the European Research Council via ERC consolidating grant 773062 (acronym O.M.J.). F.R. is supported by the Göran Gustafsson Foundation for Research in Natural Sciences and Medicine. We acknowledge support from the Swedish National Space Agency (196/16), the Swedish Research Council (Vetenskapsrådet, 2018-03513), and the Swedish Foundation for international Cooperation in Research and Higher Education (STINT, IB2019-8160).

## Author contributions

H.D.-B., A.P. and F.R. wrote the manuscript. H.D.-B. has performed sample selection and temporal analysis of the data, theoretical calculations and interpretation. A.P. provided theoretical calculations, interpretation and insight. F.R. assists in the sample selection, interpretation and insight of the results. S.-R.O. provides the *Swift*-UVOT count rate light curves and assists in the correction and conversion processes of the data. B.Z. assists in the theoretical calculations, discussion and representation of the results. M.-G.D. provides the X-ray sample of 222 GRBs and the optical sample of 102 GRBs with all required parameters and assists in the discussion of those two samples. The parameters of those samples are used for initial discussion. M.-G.D. assists in the general discussion of the paper as well as the discussion of the Fermi-LAT paper with flat phase. All authors reviewed the manuscript.

## Competing interests

The authors declare no competing interests.
