## [Peer Review File · Nature Communications]

A wind environment and Lorentz factors of tens explain gamma-ray bursts X-ray plateauEditorial Note: Parts of this peer review file have been redacted as indicated to avoid any copy right infringement.

REVIEWER COMMENTS

Reviewer #1 (Remarks to the Author):

The work by Dereli-Bégué et al. argues in favor of the standard external shock scenario for explaining the X-ray plateaus observed in the numerous GRB afterglows. To match the flat X-ray lightcurves and the joint optical data, they require quite small initial Lorentz factors of the GRB jets and uncommonly low densities of the circumburst medium. By these requirements they infer strong implications to the GRB progenitors.

I would like to raise 3 major issues with the model and the parameters they adopt:

1) If the initial Lorentz factors of jet in the afterglow phase are of the order of few (upper limits of are 2-37 for the case II in Table 7), then it is not clear how could we observe MeV photons from GRBs in the prompt phase (the old compactness problem). If the model proposed by the authors is correct, then the GRBs with the X-ray plateaus (at least fraction of them which correspond to their case II) should have very particular prompt emission spectra with the absence of the high-energy power-law segment ($> \sim 1$ MeV) and with lower energies spectral peaks. Unfortunately, the 5 GRBs (case II) reported now lack the Fermi/GBM data. Authors should select similar GRBs with the X-ray plateau and demonstrate clearly that these GRBs differ from the general population in terms of their spectral peaks and the availability of the MeV data. However, given the fact that we do observe the X-ray plateaus from SGRBs which are typically harder, the argumentation against the compactness problem would be quite challenging.

2) Authors choose a specific sample of GRBs accounting only for the X-ray plateaus which follow with a normal forward shock-like decay. Therefore, they can interpret them in the standard afterglow scenario with unusual parameters (small Lorentz factors and low ISM densities).

However, they ignore the X-ray plateaus which end with the very sharp decline (t^{-3} and much sharper, so called "internal plateaus", e.g. Troja et al. 2007). It is clear that in the forward shock scenario it wouldn't be possible to produce such steep declines.

If we look in one of the latest X-ray plateau catalog by Tan et al. 2019, we can see an order of 10 sources like that.

The authors should clearly indicate the limitations of the standard forward shock model they use. If the

authors can not explain the "internal plateaus", then they have to explain this "coincidence", i.e. how 2

different mechanisms could produce the same feature, or they should point on the distinctive differences between the "normal" and the "internal" plateaus.

3) Authors choose a range between 10^{-2} to 1 for the forward shock magnetisation parameter ϵ_B . Typically,

what we derive from the usual "non-problematic" optical-to-X-ray lightcurves is much smaller ϵ_B .

The values of ϵ_B inferred from the detailed multiwavelength afterglow modelling range from 10^{-5} to 1 or even with 10^{-5} as upper limits (e.g., Santana, Duran & Kumar 2014). The authors should

clearly show the consequences for such small ϵ_B in their cases. From their Table 7, they obtain the

upper limits on the initial Lorentz factors of order of few already for $\epsilon_B = 10^{-2}$. It would mean that

for even smaller values of the magnetisation, there wouldn't be a relativistic solution for the X-ray plateaus.

Given the abovementioned major issues, I am afraid I can not recommend this paper for the publication, unless

authors find strong arguments against the raised points and/or provide new evidences to support their idea.

Best regards,

Reviewer #2 (Remarks to the Author):

The paper propose an interesting interpretation for the X-ray plateau phase in GRBs. The key physics are a lower Lorentz factor and a lower density profile for the wind from a progenitor than the values conventionally assumed. This naturally reproduces the X-ray lightcurves. Before accepting for the publication, I have comments to be included, especially about the presentation style of this paper.

The conclusions in this paper are simple; they require a lower Lorentz factor and a weaker wind. However, the description of the contents is complicated so that it is hard to catch the contents by this short main text. There are three cases for each GRB, six regions for the spectrum, two stages in the lightcurves, two observation bands, and many parameters degenerating. I feel we need a schematic picture, from which readers catch the situation at a glance.

Figure 1 shows three cases with different colors. What is implied

by this figure denoting the three cases? There is a correlation between the case and another quantity?

Figure 2 and 3 are also busy. It is hard to distinguish the data for X-ray and optical.

In those figures, there are several samples with spectral index is close to zero. Are these consistent with a fiducial value of p ? Can we neglect such an extraordinary GRBs?

Figure 4 and 5 are also hard to understand at a glance. The numbers in the x-axis are not defined. I need the exact criterion to determine the region. It seems that authors determined the region by eye. In Figure 4, the index derived from X and optical seems inconsistent. This inconsistency is acceptable for this model?

Figure 6 is also very busy. Upper and lower limits are given. We need an explanation what case gives an upper limit rather than lower limit, and vice versa. I do not understand what blue and black lines indicate differently. The value for ϵ_B seems chosen randomly for each GRB.

In Figure 7, three cases are linked to a specific value of ϵ_B separately. What is the reason for that?

We would like to know how robust the determination of the three cases is.

The index p is determined by lightcurves only. Can we check that with the spectral data?

The lightcurve break is caused by the onset of deceleration and passing ν_c through the observation band. This paper distinguish those two cases?

What is the difference in the notations T_a and $t_{\text{ob.}}^{\text{trans}}$?

In equation (16) etc., how did the authors constrain the total energy E ?

If the wind becomes weak before the explosion, how long such a wind should continue? Is that significantly shorter than the duration of the finale evolutionary stage of progenitors?

Reviewer #3 (Remarks to the Author):

Dear Editor & Authors,

I have read the manuscript "The X-ray plateau phase of gamma-ray burst originating from an expanding shell with a Lorentz factor of a few tens" by Dr Dereli-Bégué et al. with great interest.

Gamma-ray bursts (GRBs) remain one of the most energetic explosions in the Universe, and many aspects of the progenitor and GRB physics remain obscure. The current analysis is sound, but the wider context/relevance of the results is not yet sufficiently described.

The authors find that the Lorentz factor of the jet is an order of magnitude lower than usually cited, and that the circumstellar medium (CSM) is 3-4 orders of magnitude below the anticipated Wolf-Rayet (WR) wind.

My suggestion to improve the context is to first provide an overview of the evidence to date that long-duration GRBs indeed arise from WR progenitors in the first place.

Secondly, the current status of WR winds needs to be described, both in terms of expected mass-loss rates and wind velocities.

Moreover the words "wind density" and "A_star" are referred to too loosely and imprecisely and should be discussed more clearly.

After these issues have been properly introduced (somewhere at the start of the paper), the results need to be discussed in the context of these evolutionary and CSM physical aspects. How strong is the evidence for/against WR stars as GRB progenitors?

Best Regards

REVIEWER COMMENTS AND ANSWERS

We thank the reviewers for their very useful comments, which helped us to improve the manuscript. We addressed very carefully all the reviewer's queries, as are listed below. We believe that the revised manuscript is much clearer now, and is ready for publication in Nature Communications.

Reviewer #1 (Remarks to the Author):

The work by Dereli-Bégué et al. argues in favor of the standard external shock scenario for explaining the X-ray plateaus observed in the numerous GRB afterglows. To match the flat X-ray lightcurves and the joint optical data, they require quite small initial Lorentz factors of the GRB jets and uncommonly low densities of the circumburst medium. By these requirements they infer strong implications to the GRB progenitors.

I would like to raise 3 major issues with the model and the parameters they adopt:

1) If the initial Lorentz factors of jet in the afterglow phase are of the order of few (upper limits of are 2-37 for the case II in Table 7), then it is not clear how could we observe MeV photons from GRBs in the prompt phase (the old compactness problem).

The reviewer is absolutely correct. This in fact, was one of our key motivations for the idea raised in this manuscript. When analyzing bursts with plateau, we noticed that they lack MeV emission, thereby, the compactness argument does not hold for these bursts. Indeed, no MeV photons from GRBs that we categorize as being in class II were ever reported. (Please note that in the revised manuscript, we rename the “cases I, II, III” as “classes I, II, III” to avoid any confusion with word “case” in any situation.)

In the revised version, we explicitly state that “We further emphasize that GRBs with plateau phase which have very low Lorentz factor (namely, GRBs in class II) lack any evidence of MeV emission, implying that the opacity argument cannot be used at all in these bursts.” (page 5, par. 3)

If the model proposed by the authors is correct, then the GRBs with the X-ray plateaus (at least fraction of them which correspond to their case II) should have very particular prompt emission spectra with the absence of the high-energy power-law segment ($> \sim 1$ MeV) and with lower energies spectral peaks. Unfortunately, the 5 GRBs (case II) reported now lack the Fermi/GBM data. Authors should select similar GRBs with the X-ray plateau and demonstrate clearly that these GRBs differ from the general population in terms of their spectral peaks and the availability of the MeV data.

This is a valid point. Indeed, following the reviewer's comment, we further analyzed the distribution of peak energies in bursts that (i) have plateau; (ii) do not have plateau; and (iii) have a steep slope (steeper than the limit we defined of $\alpha = 0.7$). We find that indeed the distribution of E_{peak} is different for these three classes: bursts with "plateau" are found to have a lower peak energy than other bursts.

This result is consistent with the idea that jets in bursts that show a "plateau" indeed have a lower Lorentz factor.

Our analyzed samples contained the following:

- (i) The 13 GRBs with plateau phase analyzed in this manuscript;
- (ii) 12 GRBs without plateau phase presented in Table 6 of Liang et. al. (2010). In that work, Liang et. al. estimated the Lorentz factor of these bursts using X-ray onset bump or early peak in the optical data and found that the Lorentz factor is ~few hundreds in all cases. We therefore concluded that this is a good reference for comparison the distribution of E_{peak} .

In order to ensure consistency, we did not use the values of E_{peak} as given in Liang et. al. (measured from different instruments e.g. *Konus-WIND*). Rather, we calculated E_{peak} directly from the *Swift*-BAT data. In the calculation, we used the correlation between the peak energy and the spectral index derived from fitting a single power law to a large Fermi-GBM and *Swift*-BAT data as parameterized by Virgili et al. (2012), as this method is commonly used in the literature (e.g., Zhang et al. 2007, Racusin et al. 2009). We emphasize that the result of this analysis, namely the value of E_{peak} obtained using this method is consistent with the deduced value obtained using the *Fermi*-GBM data.

In Figure 1 (here in the answer; similar to new figure 8 in the manuscript), we compare the peak energy distribution of these two samples. A clear separation is found: those GRBs which have a higher Lorentz factor indeed have a consistently higher E_{peak} than the GRBs in our sample.

In the revised manuscript, we have added this discussion in page 5, as well as in new Figure 8.

Figure 1. Distributions of peak energy. The black bars represent 13 GRBs with plateau presented in our manuscript, the red bars represent 12 GRBs without plateau phases (except GRBs 050319 and 061021 as well as early GRBs before the launch of Swift-BAT) presented in Table 6 of Liang et al. (2010). In the panel the right-hand ordinate is the number of burst in each histogram bin and the left-hand ordinate is the value of the kernel density estimation (KDE), which is shown by the black and red curves for each sample respectively. Clearly, "plateau" bursts have a consistent lower peak energy than other bursts. This result is consistent with the idea that jets in these bursts have a lower Lorentz factor.

We also performed a Kolmogorov-Smirnov test (KS test: $D = 0.60$ and $p = 0.014$) which can clearly show if these two samples originate from the same population. The KS test result shows that there is only 0.14% chance for these samples to have originated from the same population and they differ with 60%. We view this as another hint towards understanding the difference between GRBs that do show a plateau and those do not.

We next compared bursts with "flat" (temporal index $\alpha < 0.7$) and "steep: ($\alpha > 0.7$) plateaus. For that, we considered the sample of 222 GRBs with plateau phase presented in Srinivasaragavan et al. (2020), which is the most comprehensive sample of GRBs with plateau published to date. We selected 130 bursts, based on their data quality and flat X-ray plateau. Many of these bursts do not have optical data, and therefore are not part of the main analysis presented in the current manuscript.

These were compared to 33 GRBs with a steep plateau phase ($\alpha > 0.7$) taken from the same sample.

The distribution of the peak energies of both samples is presented in Figure 2 (here in the answer). As is shown in the figure, there still exists a clear separation between the distribution of E_{peak} in between the two samples. This is further

tested statistically using Kolmogorov-Smirnov (KS) test. We found that there is <1% probability that these two samples originate from the same population. It is important to mention that this sample of GRBs with “steep” X-ray plateau do not contain GRBs which do have only a single slope (self-similar slope) during their entire X-ray afterglow.

Figure 2. Distribution of peak energy for a larger sample. The black bars represent 130/222 GRBs with plateau phase, red bars represent 33/222 GRBs with plateau phase steeper than the limit (0.7) put in our manuscript. Still “plateau” bursts ($\alpha < 0.7$) have a consistent lower peak energy than other bursts ($\alpha > 0.7$). We also performed a Kolmogorov-Smirnov tests between the samples 130 GRBs and 33 GRBs (KS test: $D = 0.24$ and $p = 0.085$). The KS test result shows that there is 0.85% chance for these samples to have originated from the same population and they differ with 24%. We think that difference between two samples still holds.

In addition to the clear differences between the peak energy distribution of “plateau” and without plateau / steep plateau bursts, we also found clear differences between the high energy spectral indices of bursts in these categories.

We searched the Fermi catalogue, and found that out of 130 bursts with plateau, only 8 with a “flat” X-ray plateau ($\alpha < 0.35$) were observed by the *Fermi*-GBM – non-was detected by the *Fermi*-LAT.

Out of this sample, 6/8 were reported of having a cutoff power law as a best fit, namely, no high energy emission was detected. In the last 2 cases best fitted with power law, no emission in the BGO (>250 keV) was detected.

For comparison, of the 33 “steep” X-ray plateau bursts ($\alpha > 0.7$), 10 were observed by Fermi-GBM, and 6 of those are fitted with a smoothly broken power

law or “Band” function having beta in the range $-2.2 \dots -3$; namely, they do show evidence for a high energy emission.

To conclude, we were unable to find any GRB classified as class II that show a high-energy power-law segment ($> \sim 1$ MeV) and peak at high energies.

While we find this to be further supporting evidence, since the sample of Srinivasaragavan et al. (2020) does not contain optical data, we did not use it in the original manuscript, and we therefore reserve the discussion of these 130 bursts to a future work.

In the revised manuscript, we have mentioned this lack of the MeV emission from such GRBs in class II in the paragraph 3, page 5.

However, given the fact that we do observe the X-ray plateaus from SGRBs which are typically harder, the argumentation against the compactness problem would be quite challenging.

When analyzing the bursts presented in Srinivasaragavan et al. (2020), we found that 43/222 GRBs with plateau are classified as short GRBs. None of these bursts is classified as class II in our sample (namely, having a flat X-ray plateau and decaying optical plateau in their light curves). Furthermore, none of these sources that show flat X-ray plateau in its light curve was observed by *Fermi*-GBM.

In the revised manuscript, for this issue, we have added a new paragraph, 5th in page 5.

2) Authors choose a specific sample of GRBs accounting only for the X-ray plateaus which follow with a normal forward shock-like decay. Therefore, they can interpret them in the standard afterglow scenario with unusual parameters (small Lorentz factors and low ISM densities).

However, they ignore the X-ray plateaus which end with the very sharp decline (t^{-3} and much sharper, so called "internal plateaus", e.g. Troja et al. 2007). It is clear that in the forward shock scenario it wouldn't be possible to produce such steep declines.

If we look in one of the latest X-ray plateau catalog by Tang et al. 2019, we can see an order of 10 sources like that.

The authors should clearly indicate the limitations of the standard forward shock model they use. If the authors can not explain the "internal plateaus", then they have to explain this "coincidence", i.e. how 2 different mechanisms could produce the same feature, or they should point on the distinctive differences between the "normal" and the "internal" plateaus.

We agree that the point raised by the reviewer is a very valid one.

We first would like to stress that our selection criteria were general, as stated in the paper.

Following the referee's comment, we closely looked into this issue, and we conclude that there are two possible ways in which these bursts can be explained.

First, we think that one can interpret those sub-sample of GRBs which show a sharp decline at the end of the plateau phase in more than one way. The first point is the effect of flares on the identification of the plateau, as well as on the light curve decline. We found that Tang et. al. (2019) in their analysis, assumed that no flares exist. However, when including the effects of flares, the temporal decline index changes. This is further enhanced by the fact that in some bursts the data is very sparse – in a few cases, in fact no data exists during the plateau at all, and the existence of a plateau is inferred just by the brightness of the flare (e.g., 050730, shown below in Figure 7 and 8). We demonstrate that in Figures 5–8. On the left side we show the light curve as fitted by Tang et. al., and on the right side, the same light curves as fitted by the Swift team. Clearly, once the effects of flares are isolated, the conclusions of the two analysis are different.

As a concrete example, GRB070110 shown in Figure 5, was analyzed by Troja et. al. (2007), fitting a flat X-ray light curve up to ~16,000 s, followed by a sharp decline. However, when looking at the data, we found that flux at 8,000-16,000 s may be due to a flare, implying that the data is consistent with a single slope (no plateau phase) and a large late time flare. If this interpretation is correct, there is a single decay slope, which is naturally much shallower than the one found by Troja et. al.

A second point is that in 4 GRBs in the sample of Tang et. al., a sharp decline in the light curve is observed at times much later than the end of the plateau - after ~18,000 s, while the light curve is already during the self-similar decay phase. In these cases, at such late times, there are very few data points available. This suggests that the origin of this decay may be associated with a jet break.

Thus, overall, we conclude that the “internal plateaus” are a matter of interpretation, and more data / analysis is needed. We thus suggest here a different interpretation, which is not inconsistent with this data.

[redacted]

As an example, for different fitting/interpretation: Chincarini et al. (2007) fit the light curve to define flares as presented in Figure 7 and Tang et al. (2019) fit the same light curve to define internal plateau as presented in Figure 8.

[redacted]

Although this is our interpretation, we do acknowledge that a second possibility is that our model may not be able to explain the “internal plateau” seen in the *Swift*-BAT or *Swift*-XRT light curve of short GRBs in a few seconds (~ 180 s, Lü et al. 2015). It was proposed in the literature that such an internal plateau may be powered by a long-lasting central engine activity as the X-ray flares are powered by an extended central engine activity (Ghisellini et al. 2007). As different GRBs show a plethora of spectra and light curves, it may be reasonable to expect that a single model will not be able to explain all of the observed data. We believe that even if this is case, still our model, being able to fit many GRBs in a simple and natural way has a strong merit.

To conclude, we thank reviewer for raising this issue. We added a paragraph in the section “Comparison with ideas of explaining X-ray plateau” in pages 20 and 21 where we discuss both the above points for explaining the “internal plateau” observed.

3) Authors choose a range between $1E-2$ to 1 for the forward shock magnetisation parameter ϵ_B . Typically, what we derive from the usual "non-problematic" optical-to-X-ray lightcurves is much smaller ϵ_B . The values of ϵ_B inferred from the detailed multiwavelength afterglow modelling range from $1E-5$ to 1 or even with $1E-5$ as upper limits (e.g., Santana, Duran & Kumar 2014). The authors should clearly show the consequences for such small ϵ_B in their cases. From their Table 7,

they obtain the upper limits on the initial Lorentz factors of order of few already for $\epsilon_B=1E-2$. It would mean that for even smaller values of the magnetisation, there wouldn't be a relativistic solution for the X-ray plateaus.

We agree with the reviewer that indeed there is a large uncertainty in the value of the magnetization parameter, ϵ_B , which is not measured directly. For GRBs in our class II, an indirect constraint is obtained from the requirement $\Gamma > 1$ (see Equation 18). For GRBs in our classes I and III, indeed values of ϵ_B as low as 10^{-5} or even lower can be assumed; in these cases, the inferred ambient density is close to the expected by a Wolf-Rayet star, namely higher than we initially assumed, but may in fact be an indication that indeed the values of ϵ_B are low.

In the revised version, we have added a new paragraph (2nd in page 4) where we discuss this issue. We further modified Figure 6 to include low values of ϵ_B as suggested by the reviewer.

Given the abovementioned major issues, I am afraid I can not recommend this paper for the publication, unless authors find strong arguments against the raised points and/or provide new evidences to support their idea.

Best regards,

We very much hope that the additional evidence we provide in our answers sufficient to convince the referee that the idea we raise in this manuscript lies on a solid ground to justify its publication.

Reviewer #2 (Remarks to the Author):

The paper propose an interesting interpretation for the X-ray plateau phase in GRBs. The key physics are a lower Lorentz factor and a lower density profile for the wind from a progenitor than the values conventionally assumed. This naturally reproduces the X-ray lightcurves. Before accepting for the publication, I have comments to be included, especially about the presentation style of this paper.

The conclusions in this paper are simple; they require a lower Lorentz factor and a weaker wind. However, the description of the contents is complicated so that it is hard to catch the contents by this short main text. There are three cases for each GRB, six regions for the spectrum, two stages in the lightcurves, two observation bands, and many parameters degenerating. I feel we need a schematic picture, from which readers catch the situation at a glance.

We agree that while the theory of synchrotron emission, which we adopt, though simple and well-studied, can lead to complicated and somewhat confusing outcome. In the manuscript, we added Figure 1 (Figure 9 here below) to clarify the different cases. In addition, we also presented three examples for 3 different classes I, II, III.

Figure 9: Temporal evolution of injection frequency, ν_m (red points) and cooling frequency ν_c (blue crosses) in the coasting phase into a wind medium. The letters (A, B, C, D, E, F) represent the six possible spectral and temporal regions in Table 4 in the manuscript. Orange and blue horizontal lines represent the X-ray and optical frequencies respectively. Red dashed vertical line indicates the crossing of both injection and cooling frequencies. It is clear that a given observed frequency can shift from one region to another only along one of the following paths: i) B-C-F-E or ii) B-A-D-E.

Figure 1 shows three cases with different colors. What is implied by this figure denoting the three cases? There is a correlation between the case and another quantity?

In Figure 2 (old Figure 1) we show that the three classes (I) – (III), selected only based on the temporal indices of their X-ray and optical afterglow light curves, are, in addition, correlated with the prompt phase energy as well as the break

time. Furthermore, bursts in the three classes seem to occupy different region in the $E_{\text{iso}} - T_{\{a,X\}}$ parameter space. This fact provides a further, independent tool to increase our confidence in the selection criteria we use for these classes.

We added a few sentences in the caption of Figure 2 (old Figure 1) to explain the correlation between both quantities and the role of classes in this correlation.

Figure 2 and 3 are also busy. It is hard to distinguish the data for X-ray and optical.

We added two panels to separate X-ray and optical data in both phases (plateau and self-similar) in the same Figures 3 (old Figure 2) and 4 (old Figure 3) respectively.

In those figures, there are several samples with spectral index is close to zero. Are these consistent with a fiducial value of p ? Can we neglect such an extraordinary GRBs?

We apologize for this mistake. These bursts had no spectral index available, so we erroneously wrote them as 0. In the revised figures, we simply removed them, and added a note in the caption.

There are two cases – GRBs 060614 and 060729, for which there is an inconsistency between the power law index derived using temporal and spectral data. However, we point that the values obtained using the X-ray and optical temporal indices, and X-ray spectral indices, are all consistent with each other. Only the values obtained using the optical spectral data deviates. Since no errors are given in the literature, we cannot estimate the reliability of the optical data for these particular bursts, and we therefore accept the common value of p obtained using other 6 indices.

We added an explanation on the caption of Table 5 to clarify this issue.

Figure 4 and 5 are also hard to understand at a glance. The numbers in the x-axis are not defined. I need the exact criterion to determine the region.

First, we combined both Figures into new Figure 5. Second, we made a major change in the caption to make Figure 5 easily understood. The numbers on the x-axis are removed, as they are indeed not needed.

It seems that authors determined the region by eye. In Figure 4, the index derived from X and optical seems inconsistent. This inconsistency is acceptable for this model?

For each time interval (plateau and decaying afterglow) ideally, there are 8 independent ways of measuring the electron power law index, p , namely by using the spectral and temporal indices of both X-ray and optical bands. We find the measurements based on temporal indices to be more accurate, simply because

the spectral data is band limited, while the temporal data extends over 2-3 orders of magnitude, allowing a more accurate determination of the value of p . Having said that, in the revised Figure 5, we use both the temporal and spectral data to estimate the value of p . For consistency, we measure the value of p both during the plateau and the late time decay (attributed to the self-similar motion). In three bursts we find that 6 out of 8 indices provide consistent value of p (for a given region, either [E] or [F]), while 2 out of 8 suggests a different value. This is often the result of lack, or very weak optical data at later times, which affects the fit. Therefore, in such a case, we consider the result as suggested by the majority of independent methods as accepted.

We added an explanation on the caption of Table 5 where we provide the full explanation.

Figure 6 is also very busy. Upper and lower limits are given. We need an explanation what case gives an upper limit rather than lower limit, and vice versa. I do not understand what blue and black lines indicate differently. The value for ϵ_B seems chosen randomly for each GRB.

We thank the reviewer for pointing out all those confusing cases, we changed the colors on the plot and caption of Figure 6 to clarify. The colors now represent the 3 classes. As we explain in the theory part, only for bursts in class I we can obtain direct values of the Lorentz factor (Γ) and ambient density, for a given ϵ_B . For bursts in class II upper limits on Γ are obtained, while for bursts in class III, lower limits are obtained.

Following the reviewer's suggestion, we have chosen the same value of ϵ_B for all GRBs. Since the exact value of ϵ_B is unknown, we selected several representative values discussed in the literature ($\epsilon_B = 10^{-5}$, 10^{-3} and 10^{-2}), and added the relevant constraints.

In Figure 7, three cases are linked to a specific value of ϵ_B separately. What is the reason for that?

In the revised Figure 7, we calculated all values of the Lorentz factor for a single value of $\epsilon_B = 10^{-3}$ as a representative value. We do note the very large uncertainty in the magnetization representing a gap in our knowledge – up to several orders of magnitude (e.g., Santana, Barniol-Duran & Kumar, 2014).

In the revised manuscript, we added an explanation about this in the second paragraph, page 4.

We would like to know how robust the determination of the three cases is.

These three classes are determined based on their X-ray and optical temporal properties. class I: flat X-ray, decaying optical plateaus, class II: both flat X-ray and optical plateaus, class III: both decaying X-ray and optical plateaus. Here, “flat” we mean temporal index $\alpha < 0.35$, and “decaying” means $0.35 < \alpha < 0.7$. Steeper values of α are not considered as “plateau” here, similar to what is common in the literature (see Dainotti et al. 2016, 2020, Srinivasaragavan et al 2020). This separation is further strengthened by the differences in $E_{\text{iso}} - T_{\text{a,X}}$, as presented in Figure 2 (old Figure 1). While clearly there is some uncertainty, out of 222 bursts we selected only those 13 bursts in which the data quality (optical data) is sufficiently high to ensure robustness of the results.

We think that these two points, which are discussed in the paper, clearly indicate the robustness of the determination.

The index p is determined by lightcurves only. Can we check that with the spectral data?

In the revised manuscript, we have used both temporal and spectral data in determining the power law index, p . We find that in 9/12 GRBs the optical spectral data is consistent with the values derived using temporal data (both X-rays and optical), and in 10/13 GRBs the X-ray spectral data is consistent with the values derived from the temporal data. We believe that the most likely origin of the deviation in the other 3 cases are due to a combination of limited band, and too-simplistic model (i.e., that does not consider curvature of the spectra, etc.). In all cases, the obtained value of p is consistent with at least 6/8 independent measurements. We added some explanation in the caption of the Table 5 to explain these inconsistent cases.

The lightcurve break is caused by the onset of deceleration and passing ν_c through the observation band. This paper distinguish those two cases?

We do not make an explicit distinction, but rather assume a continuity in time.

What is the difference in the notations T_a and $t^{\text{ob.}}_{\text{trans}}$?

We apologize for this. The two are the same. This is fixed in the revised manuscript.

In equation (16) etc., how did the authors constrain the total energy E ?

The total energy (E_{iso}) is computed by using equation 1 and *Swift*-BAT data. (cf. Methods, Isotropic energy, in page 12). We added a note for E_{iso} below the equation 16 and all the other parameters e.g., T_a , ν_{Fnu_X} and ν_{Fnu_O} .

If the wind becomes weak before the explosion, how long such a wind should continue? Is that significantly shorter than the duration of the finale evolutionary stage of progenitors?

Observationally, very little is known about the final stages of the evolution of the most massive stars (luminous blue variables and Wolf-Rayet stars), of which some lead to an evolutionary channel which end up as GRBs. Rapid evolutionary stages of such stars are expected during the last 10's of centuries of their life, which will have profound effect on the circumstellar wind profiles. Instabilities will cause elevation of the outer envelope which can cause occasional giant eruption events, with major mass ejections in several consecutive periods. These mass ejections lead to circumstellar nebulae and wind-blown bubbles. Observations of galactic Wolf-Rayet stars indicate shell structures and nebulae at 1-10 pc scales, and in some cases, with low density cavities within (Toal et al 2013).

In the revised manuscript, we added a new paragraph (last of page 5/first of page 6) where we discuss the evidence for/against WR stars as GRB progenitors.

Reviewer #3 (Remarks to the Author):

Dear Editor & Authors,

I have read the manuscript "The X-ray plateau phase of gamma-ray burst originating from an expanding shell with a Lorentz factor of a few tens" by Dr Dereli-Bégué et al. with great interest.

Gamma-ray bursts (GRBs) remain one of the most energetic explosions in the Universe, and many aspects of the progenitor and GRB physics remain obscure. The current analysis is sound, but the wider context/relevance of the results is not yet sufficiently described.

The authors find that the Lorentz factor of the jet is an order of magnitude lower than usually cited, and that the circumstellar medium (CSM) is 3-4 orders of magnitude below the anticipated Wolf-Rayet (WR) wind.

My suggestion to improve the context is to first provide an overview of the evidence to date that long-duration GRBs indeed arise from WR progenitors in the first place.

Secondly, the current status of WR winds needs to be described, both in terms of expected mass-loss rates and wind velocities.

We are very thankful to the reviewer for such a suggestion. In the revised version, we added a new paragraph (second paragraph in page 1) in the main text where we clearly explain all those terms. Furthermore, we point out that the use of lower values of the magnetization parameter implies that the wind density may be somewhat higher than our initial expectation, namely 1-2 orders of magnitude below the anticipated from a WR star.

Moreover the words "wind density" and "A_star" are referred to too loosely and imprecisely and should be discussed more clearly.

In page 3, we removed the word "density" and add new information to better explain the relation between density and A_star. Moreover, we also refer to the supplementary documents.

After these issues have been properly introduced (somewhere at the start of the paper), the results need to be discussed in the context of these evolutionary and CSM physical aspects. How strong is the evidence for/against WR stars as GRB progenitors?

In the revised version, we added a new paragraph at the end of page 5 / beginning of page 6 discussing this. We show that the density (A_star) is mostly 1-2 order of magnitude smaller than the one expected from the WR stars. As we explain in the manuscript, we cannot consider this as evidence against WR progenitor stars, as little is known about the properties of the wind in the final stages (10's of centuries) of evolution of a WR star. We thus view one of the merits of this work as providing further information that could potentially help understanding the nature of these objects.

Best Regards

In addition to the answers to the reviewer comments we have made some further improvements on the manuscript, which are highlighted by bold fonts.

REVIEWER COMMENTS

Reviewer #1 (Remarks to the Author):

The revised manuscript by H. Dereli-Bégué et al. on the origin of the X-ray plateaus have been improved significantly compared to the first version.

However, I still find a major issue with the model they propose.

Given that the scenario has big impact on the progenitors of GRBs, I can not recommend the paper for the publication without a proper answer to the following point:

Authors find a statistically significant difference between the distribution of peak energies, but unfortunately authors do not discuss anywhere in the text other fundamental problems with such small bulk Lorentz factors (as small as <5 according to their Figure in the Methods for the distribution of Gamma). I can imagine that there is a special class of GRBs that have spectral cutoff in their spectra at $\sim \text{few} \times 511 \text{ keV}$, and only those GRBs have the plateau emission, but with $\Gamma=5$, one should guarantee that the GRB jet is transparent. It would have a size of $R \sim 1.5E12 \text{ cm}$ for $\Gamma=5$ and variability of 1 s (even though GRBs have $dt < 1 \text{ s}$). Given the typical energetics of GRBs, isn't this R below the photosphere? The jet with such small Γ should become transparent at $R_{\text{ph}} \sim 1E18 \text{ cm}$ or even at larger radii (Daigne and Mochkovitch 2002). At such large radii, to obtain a typical variability of GRBs, one needs to invoke either very large bulk Lorentz factor (contradiction with assumed Γ of few) or mini-jets (with $\Gamma_{\text{mini_jet}} \gg \Gamma$), but again, one needs to find a way to efficiently kill the mini-jets before the jet decelerates in the circumburst medium. To summarise my point, it is extremely difficult to build a scenario for a 1 s-scale (or more realistically sub-second) varying transient with such huge total energy to originate from a jet with such small bulk Lorentz factor. It is easy in blazars, simply because the observed variability time scale is much longer, the total energy of the jet is less, and the radii of the internal dissipation are much larger and there is no such a strong competition at those radii between the internal and the external dissipation of jet. Authors need to find a convincing self-consistent scenario for the prompt + afterglow emission with $\Gamma \sim \text{few}$.

Reviewer #2 (Remarks to the Author):

I found that the revised version satisfies all my requirements.

Reviewer #3 (Remarks to the Author):

The discussion has notably improved.

My comments have been accounted for.

I recommend publication of the MS.

REVIEWER COMMENT and ANSWER

We are very glad to see that Referees #2 and #3 are now fully convinced by our manuscript and we thank you for giving us the opportunity to complete our response to the final raised scientific query and convince Referee #1.

Reviewer #1 (Remarks to the Author):

The revised manuscript by H. Dereli-Bégué et al. on the origin of the X-ray plateaus have been improved significantly compared to the first version. However, I still find a major issue with the model they propose. Given that the scenario has big impact on the progenitors of GRBs, I can not recommend the paper for the publication without a proper answer to the following point:

Authors find a statistically significant difference between the distribution of peak energies, but unfortunately authors do not discuss anywhere in the text other fundamental problems with such small bulk Lorentz factors (as small as <5 according to their Figure in the Methods for the distribution of Gamma). I can imagine that there is a special class of GRBs that have spectral cutoff in their spectra at $\sim \text{few} \times 511 \text{ keV}$, and only those GRBs have the plateau emission, but with $\text{Gamma}=5$, one should guarantee that the GRB jet is transparent. It would have a size of $R \sim 1.5E12 \text{ cm}$ for $\text{Gamma}=5$ and variability of 1 s (even though GRB have $dt < 1 \text{ s}$). Given the typical energetics of GRBs, isn't this R below the photosphere? The jet with such small Gamma should become transparent at $R_{\text{ph}} \sim 1E18 \text{ cm}$ or even at larger radii (Daigne and Mochkovitch 2002). At such large radii, to obtain a typical variability of GRBs, one needs to invoke either very large bulk Lorentz factor (contradiction with assumed Gamma of few) or mini-jets (with $\text{Gamma}_{\text{mini_jet}} \gg \text{Gamma}$), but again, one needs to find a way to efficiently kill the mini-jets before the jet decelerates in the circumburst medium. To summarise my point, it is extremely difficult to build a scenario for a 1 s-scale (or more realistically sub-second) varying transient with such huge total energy to originate from a jet with such small bulk Lorentz factor. It is easy in blazars, simply because the observed variability time scale is much longer, the total energy of the jet is less, and the radii of the internal dissipation are much larger and there is no such a strong competition at those radii between the internal and the external dissipation of jet. Authors need to find a convincing self-consistent scenario for the prompt + afterglow emission with $\text{Gamma} \sim \text{few}$.

First, we would like to thank the referee for raising this point. We highly appreciate the physical insight of the referee, which forces us to look very carefully at all aspects of the model that we advocate here. We find this to be very constructive and help us clarify all points that seem potentially weak in our model.

The comment made by the referee is a valid one. Indeed, if the variability time is small (of the order of a second or less), and the Lorentz factor is no more than a

few, the emission radius of the prompt photons could be below the photosphere, necessitating a quasi-thermal spectrum during the prompt phase of GRBs, and smearing short variability time. For a $\Delta_t(\text{min}) = 0.1$ second variability, and isotropic luminosity $L_{\text{iso}} = 10^{50.5}$ erg/s, comparison of the estimated prompt emission radius, $R_E \sim (2 \Gamma^2 c \Delta_t(\text{min}))$ with the photospheric radius, $R_{\text{ph}} = L \sigma_T / (8 \pi m_p c^3 \Gamma^3)$ reveals that this scenario requires $\Gamma_i > \sim 30$.

In our sample, there are 4 bursts (060614, 060729, 100418A, 171205A) for which $\Gamma_i < 30$, all fall within class II. The bursts we have in this class are characterized by having (i) low luminosity: 10^{46} erg/s $< L_{\text{iso}} < 10^{50}$ erg/s, as opposed to $L_{\text{iso}} > 10^{50}$ erg/s in all other 9 bursts in our sample; and (ii) smooth prompt emission light curves, with measured variability time $\Delta_t(\text{min}) > 5$ s when such a measurement exists e.g. 060729 (Golkhou et al. 2014). [In fact, having low luminosity implies relatively low signal to noise ratio, $S/N < 10$ (Golkhou et al. 2014) which means that the standard measurement of the minimum variability timescale for these four GRBs via wavelet transform method is very difficult. Therefore, we estimated those values by using the *Swift*-BAT light curve in the online repository of NASA where the higher $S/N=5$ ratio is considered. We find that a 5 s variability is a conservative estimate, while the true variability of some of these bursts may be a few times longer. Indeed, our result is consistent with that of Sonbas et al. (2015), who showed that the low luminosity GRBs have higher minimum variability timescale than high luminosity GRBs (see their figure 4).] Combined, these observational facts imply that the constraint on the value of Γ_i is even lower, of the order ~ 10 .

The most important point, though, is not even related to the variability. As stated, all these bursts are in class II, namely, their optical light curve is flat. As we show in the theory section of the manuscript, for this class we can only constrain the quantity $(A_{\text{star}} \Gamma_i^4)$, where A_{star} is the ambient density and Γ_i is the coasting Lorentz factor (see equation 16). However, we cannot fully remove the degeneracy between the values of A_{star} and Γ_i without additional knowledge. For that we assume knowledge (or physical constraints) of the magnetization, ϵ_B (Equation 18). The method we choose is to assume a value of ϵ_B , from which this degeneracy is broken. This is plotted by the arrows (of different colors) in Figure 6 in the manuscript. Generally, the smaller the value of ϵ_B is, the denser the medium is (larger A_{star}) and simultaneously, the smaller the value of the coasting Lorentz factor we obtain.

Given the huge uncertainty in the value of ϵ_B (e.g., Santana, Barniol Duran & Kumar 2014 and Barniol Duran 2014), the only physical constraint we feel safe to put is $\epsilon_B < 1$. In the original manuscript, we cited and used average values of $\epsilon_B = 10^{-3}$. When taking higher (yet, physically acceptable) value of $\epsilon_B = 0.1$, we find that the values of Γ_i in all but one burst become ~ 10 , and fulfills the physical requirement $R_E > R_{\text{ph}}$. This criteria is also fulfilled by the last burst, GRB 171205 for which $\Gamma_i = 4$, but the luminosity

is extremely low, $L_{\text{iso}} = 5.6 \cdot 10^{46}$ erg/s. We point out that a slightly higher value of ϵ_B , say $\epsilon_B = 0.3$ further substantially increases the ratio R_E/R_{ph} due to the strong dependence on the value of the Lorentz factor.

In table 1 below we summarize the obtained prompt emission and photospheric radii of these 4 bursts, under the assumption of $\epsilon_B = 0.1$, and variability time $\Delta t(\text{min}) = 5$ s:

Table 1

GRB name	Emission Radius (R_E)	Photosphere Radius (R_{ph})	Ratio(R_E/R_{ph})
060614	$3.8 \cdot 10^{13}$	$2.8 \cdot 10^{12}$	13.7
060729	$6.6 \cdot 10^{13}$	$2.8 \cdot 10^{12}$	23.5
100418A	$2.2 \cdot 10^{13}$	$5.3 \cdot 10^{13}$	0.4
171205A	$3.9 \cdot 10^{12}$	$7.1 \cdot 10^{11}$	5.5

We thus conclude that while the point raised by the referee is a very valid one, a close look at the available data reveals that there is no inconsistency between the low values of Γ we find and the requirement that the prompt emission occurs above the photosphere.

In the revised manuscript, we made several modifications to include a complete discussion about this important point.

First, we added a new paragraph (last paragraph on page 4) where we thoroughly discuss this requirement and its implication on the value of ϵ_B for these bursts.

Second, we added a column to Table 6, where we add the data on the duration of the prompt phase (T_{90}) such as to enable the reader to calculate L_{iso} directly.

Third, we modified the caption of Table 7, where we give the values of Γ_i and A_{star} obtained for the low- Γ GRBs under the assumption of $\epsilon_B = 0.1$

Fourth, we added new arrows (blue) to Figure 6, indicating the values obtained for both Γ_i and A_{star} for $\epsilon_B = 0.1$.

Finally, we added a new figure (Figure 7 in the revised manuscript) where we plot the ratio R_E/R_{ph} , which shows directly that the prompt emission radius is always greater than the photosphere in all cases [except for GRB100418A, where they are similar for $\epsilon_B = 0.1$; For $\epsilon_B = 0.4$, this ratio will become >1 . To illustrate the uncertainty in R_E/R_{ph} due to the magnetization, we added green lines indicating this ratio for values of ϵ_B ranging from 10^{-3} [small ratio, bottom of each line] to 0.7 [large ratio].].

REVIEWERS' COMMENTS

Reviewer #1 (Remarks to the Author):

The revised version of the manuscript has significantly improved and I suggest it for the publication.

However, in my opinion, the authors should clearly state in the Abstract and in the end of the main text that their model has concrete predictions, which can be tested to verify or reject the model in the future, i.e. long variability time-scale for GRBs with a given type of a plateau emission component.

REVIEWER's COMMENT and ANSWER

We are very glad to see that Referee #1 is now fully convinced by our manuscript and purposed for publication. Following we addressed the final remark of the Referee #1.

Reviewer #1 (Remarks to the Author):

The revised version of the manuscript has significantly improved and I suggest it for the publication.

However, in my opinion, the authors should clearly state in the Abstract and in the end of the main text that their model has concrete predictions, which can be tested to verify or reject the model in the future, i.e. long variability time-scale for GRBs with a given type of a plateau emission component.

In the revised abstract, we added “We discuss several testable predictions of this model.” Due to the 150 word limitations, we were unable to further elaborate this point in the abstract. We therefore added at the end of the main text (discussion section, page 7, end of second paragraph, marked in bold face)

“Moreover, our model provides several testable predictions about bursts with long "plateau". Such bursts (i) are not expected to show high energy (>GeV) emission; (ii) are not expected to show strong thermal component; and (iii) the typical variability time during the prompt phase is expected to be long, of the order of few seconds. Exact constraints can be put on a case-by-case basis, using the equations we provide below (‘see, methods subsection Theoretical model’).”